# Homophily Enhanced Graph Domain Adaptation

**Ruiyi Fang** [1] [*]  **Bingheng Li** [2] [*]  **Jingyu Zhao** [3]  **Ruizhi Pu** [1]  **Qiuhao Zeng** [1]  **Gezheng Xu** [1]  **Charles Ling** [1] [4]
**Boyu Wang** [1] [4]

## Abstract

Graph Domain Adaptation (GDA) transfers knowledge from labeled source graphs to unlabeled target graphs, addressing the challenge of label scarcity. In this paper, we highlight the significance of graph homophily, a pivotal factor for graph domain alignment, which, however, has long been overlooked in existing approaches. Specifically, our analysis first reveals that homophily discrepancies exist in benchmarks. Moreover, we also show that homophily discrepancies degrade GDA performance from both empirical and theoretical aspects, which further underscores the importance of homophily alignment in GDA. Inspired by this finding, we propose a novel homophily alignment algorithm that employs mixed filters to smooth graph signals, thereby effectively capturing and mitigating homophily discrepancies between graphs. Experimental results on a variety of benchmarks verify the effectiveness of our method.

## 1. Introduction

In the era of massive graph data collection, graphs often integrate both structural topology and node attributes, providing rich contexts for numerous real-world applications. However, they are often constrained by label scarcity, as annotating structured data remains challenging and costly (Xu et al., 2022; 2024b; Zeng et al., 2024; 2023). To address this challenge, Graph Domain Adaptation (GDA) has emerged as an effective paradigm to transfer knowledge from labeled source graphs to unlabeled target graphs (Chen et al., 2019; Shi et al., 2024). Conventional GDA methods primarily focus on aligning graph structures across domains

by leveraging deep domain adaptation techniques (Shen et al., 2020b; Wu et al., 2020). While these approaches have achieved promising results in GDA (Yan & Wang, 2020; Shen et al., 2023), their efforts are centered on mitigating the discrepancy in structural and attribute distributions.

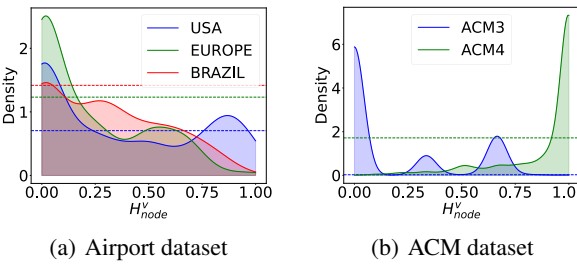

(a) Airport dataset       (b) ACM dataset

*Figure 1.* This represents node homophily distibution in two benchmark. The dotted line represents the overall node homophily ratio of the entire graph. This shows the local node homophily distibution shift is existing in various levels of homophilic groups. Homophily means that similar nodes are prone to connect to each other.

In this work, we highlight the role of graph homophily discrepancies and their impact on GDA performance. Specifically, we investigate the local homophily distribution of graph on two GDA benchmarks, as shown in Figure 1. It can be observed that while the overall homophily distribution discrepancy of the graph is relatively small, the discrepancy within homophilic and heterophilic subgroups is more substantial. For example, as shown in Figure 1 (a), the overall graph node homophily ratio between ACM3 and ACM4 appears relatively similar, and the distribution of homophilic and heterophilic groups varies more substantially. To further investigate its impact on GDA, we conduct another experiment to explore how the homophilic ratio affects the cross-network performances, as shown in Figure 2. It can be observed that the classification accuracy of target graph nodes exhibits a negative correlation with homophily divergence across different homophily ratios in datasets, which indicates the importance of separately dealing with heterophilic and homophilic groups. Besides, this observation also suggests that the discrepancies in homophilic distributions between source and target graphs may significantly impact target node classification perfor-

---
[*]Equal contribution [1]Western University, Ontario, Canada [2]Michigan State University, Michigan, USA [3]University of Electronic Science and Technology of China, Chengdu, Sichuan Province, China [4]Vector Institute, Ontario, Canada. Correspondence to: Boyu Wang <bwang@csd.uwo.ca>.

*Proceedings of the 42nd International Conference on Machine Learning*, Vancouver, Canada. PMLR 267, 2025. Copyright 2025 by the author(s).

mance, thereby highlighting the importance of mitigating homophily divergence.

To further justify this implication, we theoretically investigate the generalization performance of GDA from the PAC-Bayes perspective. Specifically, we show that the target loss can be upper bounded in terms of homophilic signal shift, heterophilic signal shift, attribute signal shift and graph node heterophily distribution shift. This aligns with our empirical findings, suggesting that discrepancies in graph signals should be addressed separately. To this end, we propose a novel cross-channel graph homophily enhanced alignment (HGDA) algorithm for cross-network node classification, as shown in Figure 3. HGDA utilizes homophilic, full-pass, and heterophilic filters to separately extract and align the homophily signal, heterophily signal, and attribute signal for both the source and target domains.

Our contributions are summarized as follows:

- Empirical insights: We first highlight the significance of homophily distribution discrepancy in GDA and empirically examine its impact on GDA performance.

- Theoretical justification: We theoretically justify the impact of homophily distribution shift on GDA and demonstrate that this discrepancy can be mitigated by addressing homophilic and heterophilic groups separately.

- Algorithmic framework: Inspired by theoretical insights, we propose HGDA, which mitigates homophily distribution shifts by aligning and capturing homophily, heterophily, and attribute signals.

## 2. Related Work

### 2.1. Heterophilic Graph Learning

Heterophilic structure is prevalent in practice, from personal relationships in daily life to chemical and molecular scientific study (Fang et al., 2022; He et al., 2025). Developing powerful heterophilic GNN models is a hot research topic. (Lim et al., 2021; Qian et al., 2024; Yang et al., 2024a; 2023b; Zhuo et al., 2024c; Li et al., 2025b) provide general benchmarks for heterophilic graph learning. In addition, many methods have been proposed to revise GNNs for heterophilic graphs. (Yang et al., 2021) specifies propagation weight for each attribute to make GNNs fit heterophilic graphs and (Li et al., 2022a) explores the underlying homophilic information by capturing the global correlation of nodes. (Zhu et al., 2020a) enlarges receptive field via exploring high-order structure. (Chien et al., 2021) adaptively combines the representation of each layer and (Chen et al., 2020) integrates embeddings from different depths with residual operation. Recent studies (Mao et al., 2024a;

Liu et al., 2025; Li et al., 2024a; 2025a; Luan et al., 2022; Huang et al., 2023; 2024; Zhuo et al., 2025; 2024b) reveal that graph homophilc and heterophilic patterns impact graph clustering performance between the test and training sets. However, the above methods failed to explore the role of homophily in GDA and minimize their discrepancy in many aspects (Kang et al., 2024; Pu et al., 2025; Xu et al., 2025; Xie et al., 2024; 2025).

### 2.2. Graph Domain Adaptation

Recent Domain Adaptation works have differences from GDA methods (Li et al., 2024b; Chen et al., 2024; Li et al., 2025c; 2024c; 2025b; Zeng et al., 2025; Xu et al., 2024a). Identifying the differences between the target and source graphs in GDA is crucial. For graph-structured data, several studies have explored cross-graph knowledge transfer using graph domain adaptation (GDA) methods (Shen & Chung, 2019; Dai et al., 2022; Shi et al., 2024). Some graph information alignment-based methods (Shen et al., 2020a;b; Yan & Wang, 2020; Shen et al., 2023; Yang et al., 2023a) adapt graph source node label information by integrating global and local structures from both nodes and their neighbors. UDAGCN (Wu et al., 2020) introduces a dual graph convolutional network that captures both local and global knowledge, adapting it through adversarial training. Furthermore, ASN and GraphAE (Zhang et al., 2021; Guo et al., 2022) consider extracting and aligning graph specific information like node degree and edge shift, enabling the extraction of shared features across networks. SOGA (Mao et al., 2024b) is the first to incorporate discriminability by promoting structural consistency between target nodes of the same class, specifically for source-free domain adaptation (SFDA) on graphs. SpecReg (You et al., 2022) applies an optimal transport-based GDA bound and demonstrates that revising the Lipschitz constant of GNNs can enhance performance through spectral smoothness and maximum frequency response. JHGDA (Shi et al., 2023) tackles hierarchical graph structure shifts by aggregating domain discrepancies across all hierarchy levels to provide a comprehensive discrepancy measure. ALEX (Yuan et al., 2023) creates a label-shift-enhanced augmented graph view using a low-rank adjacency matrix obtained through singular value decomposition, guided by a contrasting loss function. SGDA (Qiao et al., 2023) incorporates trainable perturbations (adaptive shift parameters) into embeddings via adversarial learning, enhancing source graphs and minimizing marginal shifts. PA (Liu et al., 2024c) mitigates structural and label shifts by recalibrating edge weights to adjust the influence among neighboring nodes, addressing conditional structure shifts effectively.

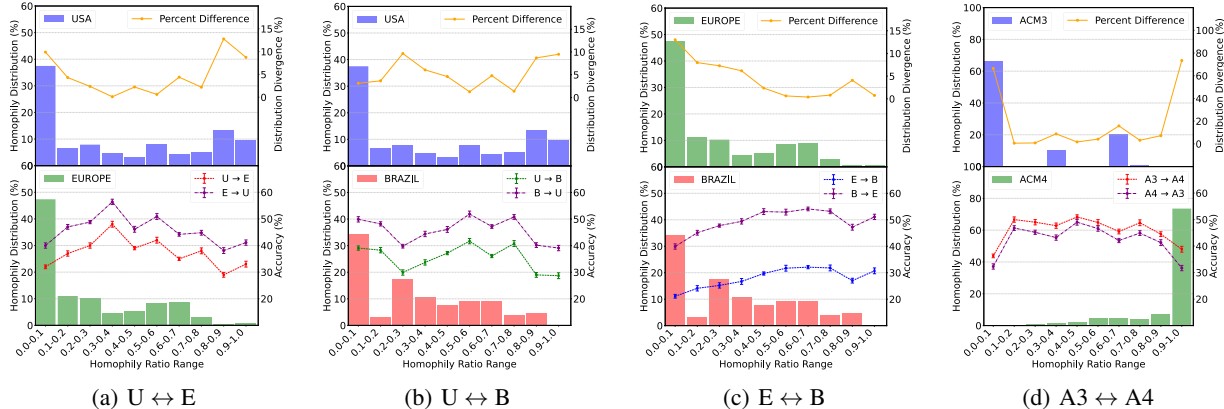

*Figure 2.* Performance of cross-network classification tasks in different homophilic ratio ranges. For a fair comparison, we use a two-layer GCN with standard unsupervised GDA settings as our evaluation model. The **X-axis** represents node subgroups categorized by specific ranges of homophily ratios. The bars in the vertical bar chart represent the proportion of nodes in each subgroup, defined by a specific homophily ratio range relative to the total number of nodes in the graph (left **Y-axis**). The line chart in the figure above represents the difference in the proportion of nodes in each subgroup between the corresponding upper and lower graphs (upper right **Y-axis**). The line graph in the figure below represents the target node classification accuracy for two corresponding tasks in each subgroup with different homophily ratios (lower right **Y-axis**). We observe that homophily divergence has a negative correlation with target node classification accuracy across various homophily ratios.

## 3. Empirical and Theoretical Study

### 3.1. Notation

An undirected graph $G = \{\mathcal{V}, \mathcal{E}, A, X, Y\}$ consists of a set of nodes $\mathcal{V}$ and edges $\mathcal{E}$, along with an adjacency matrix $A$, a feature matrix $X$, and a label matrix $Y$. The adjacency matrix $A \in \mathbb{R}^{N \times N}$ encodes the connections between $N$ nodes, where $A_{ij} = 1$ indicates an edge between nodes $i$ and $j$, and $A_{ij} = 0$ means the nodes are not connected. The feature matrix $X \in \mathbb{R}^{N \times d}$ represents the node features, with each node described by a $d$-dimensional feature vector. Finally, $Y \in \mathbb{R}^{N \times C}$ contains the labels for the $N$ nodes, where each node is classified into one of $C$ classes. Thus, the symmetric normalized adjacency matrix is $\tilde{A} = (D + I)^{-\frac{1}{2}}(A + I)(D + I)^{-\frac{1}{2}}$ and the normalized Laplacian matrix is $\tilde{L} = I - \tilde{A}$. In this work, we explore the task of node classification in a unsupervised setting, where both the node feature matrix $X$ and the graph structure $A$ are given before learning. Now we can define source graph $G^S = \{\mathcal{V}^S, \mathcal{E}^S, A^S, X^S, Y^S\}$ and target graph $G^T = \{\mathcal{V}^T, \mathcal{E}^T, A^T, X^T\}$.

**Local node homophily ratio** is a widely used metric for quantifying homophilic and heterophilic patterns. It is defined as the proportion of a node's neighbors that share the same label as the node (Pei et al.; Zhu et al., 2020b; Li et al., 2022b; Miao et al., 2024). It is formally defined as

$$H^v_{\text{node}} = \frac{|\{u \mid u \in N_v, y_u = y_v\}|}{|N_v|} \quad (1)$$

where where $N_v$ denotes the set of one-hop neighbors of

node $v$ and $y_i$ is the node $i$ label.

### 3.2. Empirical Insight

To futher explore node homophily ratio distribution shifts on the aforementioned datasets in Figure 1, we evaluate the effectiveness of GDA on different homophily subgroups. To evaluate the homophily divergence effect to GDA, we conduct an examination of node subgroups with different homophilic and heterophilic groups. The following experiments are conducted on two common graph benchmark airport datasets with unsupervised GDA node classsification task. Specifically, we focus on how the homophily discrepancy between the source and target graphs affects the performance of node classification on the target graph. Figure 2 shows the classification accuracy for different homophily ratio subgroups. Experimental results on four datasets are presented in Figure 2. It can be observed that the classification accuracy of target graph nodes negatively correlates with homophily divergence across heterophilic and homophilic groups in the datasets. For example, as shown in Figure 2 (d), subgroups within homophily divergence in the 0.0–0.1 (heterophilic groups) and 0.9–1.0 (homophilic groups) ranges exhibit high discrepancies in terms of difference in the proportion (orange line), which leads to low node classification accuracies on the corresponding target subgroups (red and purple lines). Our observations suggest that the homophily discrepancy between the source and target graphs in corresponding subgroups has a significant impact on GDA performance. This highlights the importance of mitigating the homophily divergence in both homophilic and

heterophilic groups for GDA. Additional results on more datasets are shown in Appendix C.

### 3.3. Theoretical Analysis

In this subsection, we will propose theoretical evidence based on the PAC-Baysian framework to verify the homophily divergence effect on GDA performance and demonstrate the effectiveness of our proposed method. To effectively evaluate the performance of a deterministic GDA classifier on structured graph data, we introduce the notion of margin loss.

**Margin loss on each domain.** We can define the empirical and expected margin loss of a classifier $\phi \in \Phi$ on source graph $G^S$ and target graph $G^T$. Given $Y^S$, the empirical margin loss of $\phi$ on $G^S$ for a margin $\gamma \geq 0$ is defined as:

$$
\widehat{\mathcal{L}}_S^\gamma(\phi) := \frac{1}{N_S} \sum_{i \in \mathcal{V}^S}
$$
$$
\mathbb{1}\left[\phi_i(X^S, G^S)[Y_i] \leq \gamma + \max_{c \neq Y_i} \phi_i(X^S, G^S)[c]\right].
$$
(2)

where $\mathbb{1}[\cdot]$ is the indicator function, $c$ represents node labeling . The expected margin loss is then defined as

$$
\mathcal{L}_S^\gamma(\phi) := \mathbb{E}_{Y_i \sim \Pr(Y|Z_i), i \in \mathcal{V}^S} \widehat{\mathcal{L}}_S^\gamma(\phi) \tag{3}
$$

**Definition 1** (Graph-Level Node Heterophily Distribution). *Given a graph $G$, and for any node $v \in \mathcal{V}$, the node-level heterophily is defined as $h_G(v) = \frac{1}{|\mathcal{N}(v)|} \sum_{u \in \mathcal{N}(v)} \mathbb{I}(Y_u \neq Y_v)$, where $\mathcal{N}(v)$ denotes the set of neighbors of $v$. The graph-level heterophily distribution is then defined as the empirical distribution of the random variable $h_G(v)$:*

$$
P_G^H(h) = \frac{|\{v \in \mathcal{V} \mid h_G(v) = h\}|}{|\mathcal{V}|}, \quad h \in [0, 1]. \tag{4}
$$

*Here, $P_G^H(h)$ represents the proportion of nodes in $G$ that have a heterophily value of $h$, forming a probability distribution over possible node heterophily values.*

For a source domain $G^S$ and a target domain $G^T$, we denote their respective graph-level heterophily distribution as $P_S^H(h)$ and $P_T^H(h)$.

**Definition 2** (Kullback-Leibler Divergence). *For continuous probability distributions with probability density functions $p(x)$ and $q(x)$, the KL divergence is given by:*

$$
D_{KL}(P\|Q) = \int_{-\infty}^{\infty} p(x) \log \frac{p(x)}{q(x)} \, dx. \tag{5}
$$

*KL divergence measures the relative entropy or information loss when using $Q$ to approximate $P$. It is always non-negative, and $D_{KL}(P\|Q) = 0$ if and only if $P = Q$.*

**Definition 3** (KL Divergence Between Graph Heterophily Distributions). *Let $G^S$ and $G^T$ be two graphs from different domains with heterophily distributions: $P_S^H(h), P_T^H(h) > 0$ for all $h$ . The Kullback-Leibler (KL) divergence between $P_S^H$ and $P_T^H$ is defined as:*

$$
D_{KL}(P_S^H\|P_T^H) = \sum_{h \in \mathcal{H}} P_S^H(h) \log \frac{P_S^H(h)}{P_T^H(h)}, \tag{6}
$$

*where $\mathcal{H}$ is the set of possible node heterophily values.*

**Theorem 1** (Domain Adaptation Bound for Deterministic Classifiers). *Let $\Phi$ be a family of classification functions. For any classifier $\phi$ in $\Phi$, and for any parameters $\lambda > 0$ and $\gamma \geq 0$, consider any prior distribution $P$ over $\Phi$ that is independent of the training data $\mathcal{V}^S$. With a probability of at least $1 - \delta$ over the sample $Y^S$, for any distribution $Q$ on $\Phi$ such that the following inequality holds:*

$$
\mathcal{L}_T^0(\phi) \leq \widehat{\mathcal{L}}_S^\gamma(\phi) + \frac{1}{\lambda}\left[2(D_{\mathrm{KL}}(Q\|P) + 1) + \ln\frac{1}{\delta}\right] + \frac{\lambda}{4N_S}
$$
$$
+ \frac{\lambda \rho C C_0}{\sqrt{2\pi}\sigma \cdot N_S \cdot N_T}\left[\sqrt{D_{KL}(A^S X^S\|A^T X^T)}\right.
$$
$$
+ \sqrt{D_{KL}(X^S\|X^T)} + \sqrt{D_{KL}(L^S X^S\|L^T X^T)}
$$
$$
\left. + \sqrt{D_{KL}(P_S^H\|P_T^H)}\right]. \tag{7}
$$

where $f$ is the unspecified operator of first-order aggregation, $C$ is the number of classes, and $\rho, \sigma$ are constants controlled by the node feature distribution of different classes. Proof details and additional analysis are in Appendix A.

**How could theory further drive practice?** In Theorem 1 reveals four critical factors that may affect the GDA performance. **(I)** $D_{\mathrm{KL}}(A^S X^S\|A^T X^T)$ represents the graph homophily signal, capturing graph attributes modulated by the adjacency matrix between the source and target graphs. **(II)** $D_{\mathrm{KL}}(X^S\|X^T)$ represents the divergence in graph attribute distributions between the source and target. **(III)** $D_{\mathrm{KL}}(L^S X^S\|L^T X^T)$ represents the distribution divergence of graph heterophilc signal between the source and target graphs, where the graph attribute signal is modulated by the graph Laplacian matrix indicating. **(IV)** $D_{\mathrm{KL}}(P_S^H\|P_T^H)$ is a fixed intrinsic graph parameter that quantifies the heterophily distribution shift between the source and target graphs. In other words, to effectively align source and target graphs, one should consider the homophily, attribute, and heterophily signals.

## 4. Methology

Motivated by the aforementioned analysis, our framework needs to capture $AX$, $X$ and $LX$ of both source and target

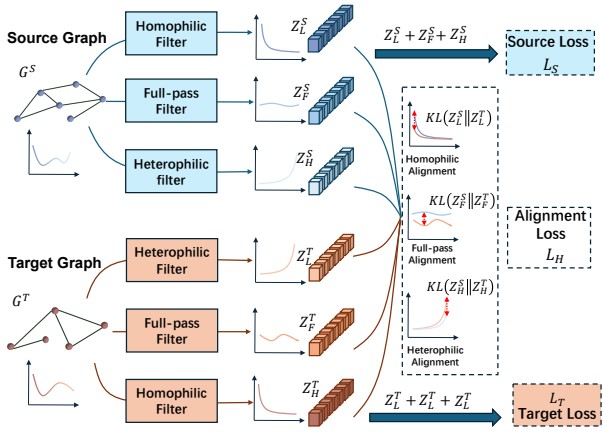

*Figure 3.* An overview of HGDA. HGDA optimizes heterophily, attribute, and homophily graph signals while minimizing homophily distribution shifts between the source and target graphs.

graph. In practice, we can treat $A$ as homophilc filter and $L$ as heterophilic filter (Nt & Maehara, 2019). Thus, as shown in Figure 3, we introduce a mixed filter to effectively capture information across different homophilic groups while minimizing various levels homophily shift.

### 4.1. Homophily, Full-pass and Heterophily Filter

**Homophilic Filter** To extract meaningful features from graphs, we utilize $H_L$ that can capture homophilc node pattens information. With the input graph $G$, the $l$-th layer's output $H_L^l$ can be represented as:

$$H_L^l = ReLU\left(\alpha^l \cdot \tilde{A}H^{l-1}W_L^{l-1}\right) \qquad (8)$$

where $H_L H^{l-1} W_L^{l-1}$ applies homophilic filtering to the node features from the previous layer $H^{l-1}$, weight matrix $W_L^{l-1}$ is a layer-specific trainable weight matrix, $\alpha^l$ is a learnable scalar parameter for the homophilic filtering, $H_L^l$ is the activation matrix in the $l$-th layer and $H^0 = X$. After applying the homophily filter to capture homophilc node signal, the output node embedding for the source graph is denoted as $Z_L^S$. Similarly, the output target homophilc node embedding is denoted as $Z_L^T$.

**Full-pass Filter** To obtain comprehensive graph attribute information, we introduce a full-pass filter to extract node attributes. In practice, the full-pass filter is defined as $H_F = \tilde{A} + \tilde{L} = I$, which is consistent with capturing all attribute information. With the input graph $G$, the $l$-th layer's output $H_F^l$ can be represented as:

$$H_F^l = ReLU\left(\alpha^f \cdot IH^{l-1}W_F^{l-1}\right) \qquad (9)$$

where $H_F H^{l-1} W_F^{l-1}$ applies full-pass filtering to the node features from the previous layer $H^{l-1}$, weight matrix $W_F^{l-1}$

is a layer-specific trainable weight matrix, $\alpha^f$ is a learnable scalar parameter for the full-pass filtering, $H_F^l$ is the activation matrix in the $l$-th layer and $H^0 = X$. After applying the full-pass filter to extract attribute information, the output node embedding for the source graph is denoted as $Z_F^S$ and output target attribute node embedding is denoted as $Z_F^T$.

**Heterophilic Filter** Similarly, we utilize $H_H$ that can capture heterophilic node pattens information. With the input graph $G$, the $l$-th layer's output $H_L^l$ can be represented as:

$$H_H^l = ReLU\left(\alpha^h \cdot \tilde{L}H^{l-1}W_H^{l-1}\right) \qquad (10)$$

where $H_H H^{l-1} W_H^{l-1}$ applies heterophilic filtering to the node features from the previous layer $H^{l-1}$, weight matrix $W_H^{l-1}$ is a layer-specific trainable weight matrix, $\alpha^h$ is a learnable scalar parameter for the heterophilic filtering, $H_H^l$ is the activation matrix in the $l$-th layer and $H^0 = X$. After applying the heterophilic filter to extract heterophilic signal, the output node embedding for the source graph is denoted as $Z_H^S$ and output target node embedding is denoted as $Z_H^T$.

### 4.2. Homophily Alignemnt Loss

The proposed framework follows the transfer learning paradigm, where the model minimizes the divergence of the two graphs. The homophily alignment method captures and aligns various homophily signals, improving target node classification performance. According to Theorem 7, the discrepancy between the source and target graphs is bounded by KL divergence. To this end, $\mathcal{L}_{\mathcal{H}}$ utilizes the KL divergence loss between the source graph embeddings $Z_L^S$, $Z_F^S$ and $Z_H^S$ and the target graph $Z_L^T$, $Z_F^T$ and the target graph embeddings $Z_H^T$, which can be formulated as:

$$\mathcal{L}_H = KL\big(Z_L^S\|Z_L^T\big) + KL\big(Z_H^S\|Z_H^T\big) + KL\big(Z_F^S\|Z_F^T\big), \qquad (11)$$

We adapt the embeddings after applying the three filters to enforce graph node alignment across different homophily ratios. Specifically, $KL(Z_L^S\|Z_L^T)$ aligns the homophilic signal, corresponding to the term $D_{\text{KL}}(A^S X^S\|A^T X^T)$. Similarly, $KL(Z_H^S\|Z_H^T)$ directly aligns the graph attributes, corresponding to the term $D_{\text{KL}}(X^S\|X^T)$. Finally, $KL(Z_F^S\|Z_F^T)$ aligns the heterophilic signal, which is consistent with the term $D_{\text{KL}}(L^S X^S\|L^T X^T)$ in Theorem 7.

### 4.3. Target Node Classification

The source classifier loss $\mathcal{L}_S\left(f_S\left(Z^S\right), Y^S\right)$ is to minimize the cross-entropy for the labeled data node in the source domain:

$$\mathcal{L}_S\left(f_S\left(Z^S\right), Y^S\right) = -\frac{1}{N_S}\sum_{i=1}^{N_S} y_i^S \log\left(\hat{y}_i^S\right) \qquad (12)$$

where $Z^S = Z_L^S + Z_H^S + Z_F^S$, $y_i^S$ denotes the label of the $i$-th node in the source domain and $\hat{y}_i^S$ are the classification

prediction for the $i$-th source graph labeled node $v_i^S \in \mathcal{V}^S$.

To utilize the data in the target domain, we use entropy loss for the target classifier $f_T$:

$$\mathcal{L}_T \left( f_T \left( Z^T \right) \right) = -\frac{1}{N_T} \sum_{i=1}^{N_T} \hat{y}_i^T \log \left( \hat{y}_i^T \right) \qquad (13)$$

where $Z^T = Z_L^T + Z_H^T + Z_F^T$, $\hat{y}_i^T$ are the classification prediction for the $i$-th node in the target graph $v_i^T$. Finally, by combining $\mathcal{L}_H$, $\mathcal{L}_S$, $\mathcal{L}_D$ and $\mathcal{L}_T$, the overall loss function of our model can be represented as:

$$\mathcal{L} = \mathcal{L}_H + \alpha \mathcal{L}_S + \beta \mathcal{L}_T \qquad (14)$$

where $\alpha$ and $\beta$ are trade-off hyper-parameters. The parameters of the framework are updated via backpropagation. A detailed description of our algorithm is provided in Appendix 14.

## 5. Experiment

We evaluate three variants of Homophily Alignemnt to understand how its different components deal with the homophilic distribution shift on 5 real-word GDA benchmarks. These variants include $\mathbf{HGDA}_L$, which uses only the homophilic filter to obtain and align node embeddings, specifically addressing distribution shifts in homophilic groups. Similarly, $\mathbf{HGDA}_F$ and $\mathbf{HGDA}_H$ only utlize the full-pass filter and heterophilic filter to obtain and align node embeddings. The detailed experimental setup can be found in Appendix B.

### 5.1. Datasets

To demonstrate the effectiveness of our approach on domain adaptation node classification tasks, we evaluate it on four types of datasets, including Airport (Ribeiro et al., 2017), Citation (Wu et al., 2020), Social (Liu et al., 2024a), ACM (Shen et al., 2024), and MAG datasets (Wang et al., 2020). The Airport dataset represents airport networks from three countries and regions: the USA (U), Brazil (B), and Europe (E). In this dataset, nodes correspond to airports, and edges denote flight routes. The Citation dataset comprises three citation networks: DBLPv8 (D), ACMv9 (A), and Citationv2 (C), where nodes represent articles and edges indicate citation relationships. As for social networks, we choose Twitch gamer networks, which are collected from Germany(DE), England(EN) and France(FR). We also use two blog networks, Blog1 (B1) and Blog2 (B2), both extracted from the BlogCatalog dataset. To further evaluate the significance of the homophily distribution effect, we curate a real-world dataset with a significant homophily distribution shift. We utilize two commonly referenced ACM datasets. The dataset ACM3(A3) is derived from the ACM

Paper-Subject-Paper (PSP) network (Fan et al., 2020), while ACM4(A4) is extracted from the ACM2 Paper-Author-Paper (PAP) network (Fu et al., 2020). These datasets inherently differ in their distributions, making them suitable for evaluating domain adaptation. For a comprehensive overview, refer to Appendix Table 6. We also provide other datasets homophily distibution in Appendix C Figure 7.

### 5.2. Baselines

We choose some representative methods to compare. GCN (Kipf & Welling, 2016) further solves the efficiency problem by introducing first-order approximation of ChebNet. DANN (Ganin et al., 2016) use a 2-layer perceptron to provide features and a gradient reverse layer (GRL) to learn node embeddings for domain classification. DANE (Zhang et al., 2019) shared distributions embedded space on different networks and further aligned them through adversarial learning regularization. UDAGCN (Wu et al., 2020) is a dual graph convolutional network component learning framework for unsupervised GDA, which captures knowledge from local and global levels to adapt it by adversarial training. ASN (Zhang et al., 2021) use the domain-specific features in the network to extract the domain-invariant shared features across networks. EGI (Zhu et al., 2021) through Ego-Graph Information maximization to analyze structure-relevant transferability regarding the difference between source-target graph. GRADE-N (Wu et al., 2023) propose a graph subtree discrepancy to measure the graph distribution shift between source and target graphs. JHGDA (Shi et al., 2023) explore information from different levels of network hierarchy by hierarchical pooling model. SpecReg (You et al., 2022) achieve improving performance regularization inspired by cross-pollinating between the optimal transport DA and graph filter theories. PA (Liu et al., 2024c) counter graph structure shift by mitigating conditional structure shift and label shift by using edge weights to recalibrate the influence among neighboring nodes. The aforementioned work does not investigate the impact of homophily distribution shift, which is an essential graph property on GDA.

### 5.3. Performance Comparison

The results of experiments are summarized in Table 1, Table 2 and Table 3, where highest scores are highlighted in **bold**, and the second-highest scores are underlined. Some results are directly taken from (Shi et al., 2023; Pang et al., 2023; Liu et al., 2024b; Zhang et al., 2025). We have the following findings: It can be seen that our proposed method boosts the performance of SOTA methods across most evaluation metrics on four group datasets, which proves its effectiveness. **HGDA** outperforms other optimal methods, achieving state-of-the-art (SOTA) performance across all datasets. Specifically, **HGDA** achieves an average improvement of $1.10\%$

| Methods | U → B | U → E | B → U | B → E | E → U | E → B | A3 → A4 | A4 → A3 | B1 → B2 | B2 → B1 |
|---|---|---|---|---|---|---|---|---|---|---|
| GCN | 0.366 | 0.371 | 0.491 | 0.452 | 0.439 | 0.298 | 0.373 | 0.323 | 0.408 | 0.451 |
| DANN | 0.501 | 0.386 | 0.402 | 0.350 | 0.436 | 0.538 | 0.362 | 0.325 | 0.409 | 0.419 |
| DANE | 0.531 | 0.472 | 0.491 | 0.489 | 0.461 | 0.520 | 0.392 | 0.404 | 0.464 | 0.423 |
| UDAGCN | 0.607 | 0.488 | 0.497 | 0.510 | 0.434 | 0.477 | 0.404 | 0.380 | 0.471 | 0.468 |
| ASN | 0.519 | 0.469 | 0.498 | 0.494 | 0.466 | 0.595 | 0.418 | 0.409 | 0.732 | 0.524 |
| EGI | 0.523 | 0.451 | 0.417 | 0.454 | 0.452 | 0.588 | 0.511 | 0.449 | 0.494 | 0.516 |
| GRADE-N | 0.550 | 0.457 | 0.497 | 0.506 | 0.463 | 0.588 | 0.449 | 0.461 | 0.567 | 0.541 |
| JHGDA | 0.695 | 0.519 | 0.511 | 0.569 | 0.522 | **0.740** | 0.516 | 0.537 | 0.619 | 0.643 |
| SpecReg | 0.481 | 0.487 | 0.513 | 0.546 | 0.436 | 0.527 | 0.526 | 0.518 | 0.661 | 0.631 |
| PA | 0.621 | 0.547 | 0.543 | 0.516 | 0.506 | 0.670 | 0.619 | 0.610 | 0.662 | 0.654 |
| **HGDA**$_L$ | 0.683 | 0.547 | 0.533 | 0.496 | 0.547 | 0.687 | 0.709 | 0.683 | 0.651 | 0.647 |
| **HGDA**$_F$ | 0.709 | 0.560 | 0.541 | 0.538 | 0.550 | 0.691 | 0.701 | 0.660 | 0.665 | 0.655 |
| **HGDA**$_H$ | 0.714 | 0.538 | 0.524 | 0.569 | 0.545 | 0.690 | 0.713 | 0.679 | 0.656 | 0.664 |
| **HGDA** | **0.721** | **0.572** | **0.569** | **0.584** | **0.570** | 0.721 | **0.718** | **0.698** | **0.683** | **0.677** |

*Table 1.* Cross-network node classification on the Airport, ACM and Blog network.

| Methods | A → D | D → A | A → C | C → A | C → D | D → C | DE → EN | EN → DE | DE → FR | FR → EN |
|---|---|---|---|---|---|---|---|---|---|---|
| GCN | 0.632 | 0.578 | 0.675 | 0.635 | 0.666 | 0.654 | 0.523 | 0.534 | 0.514 | 0.568 |
| DANN | 0.488 | 0.436 | 0.520 | 0.518 | 0.518 | 0.465 | 0.512 | 0.528 | 0.581 | 0.562 |
| DANE | 0.664 | 0.619 | 0.642 | 0.653 | 0.661 | 0.709 | 0.642 | 0.644 | 0.591 | 0.574 |
| UDAGCN | 0.684 | 0.623 | 0.728 | 0.663 | 0.712 | 0.645 | 0.624 | 0.660 | 0.545 | 0.565 |
| ASN | 0.729 | 0.723 | 0.752 | 0.678 | 0.752 | 0.754 | 0.550 | 0.679 | 0.517 | 0.530 |
| EGI | 0.647 | 0.557 | 0.676 | 0.598 | 0.662 | 0.652 | 0.681 | 0.589 | 0.537 | 0.551 |
| GRADE-N | 0.701 | 0.660 | 0.736 | 0.687 | 0.722 | 0.687 | 0.749 | 0.661 | 0.576 | 0.565 |
| JHGDA | 0.755 | 0.737 | 0.814 | 0.756 | 0.762 | 0.794 | 0.766 | 0.737 | 0.590 | 0.539 |
| SpecReg | 0.762 | 0.654 | 0.753 | 0.680 | 0.768 | 0.727 | 0.719 | 0.705 | 0.545 | 0.555 |
| PA | 0.752 | 0.751 | 0.804 | 0.768 | 0.755 | 0.780 | 0.677 | 0.760 | 0.521 | 0.538 |
| **HGDA**$_L$ | 0.756 | 0.739 | 0.794 | 0.770 | 0.739 | 0.757 | 0.765 | 0.747 | 0.532 | 0.548 |
| **HGDA**$_F$ | 0.769 | 0.747 | 0.811 | 0.758 | 0.759 | 0.782 | 0.751 | 0.749 | 0.560 | 0.539 |
| **HGDA**$_H$ | 0.752 | 0.738 | 0.804 | 0.767 | 0.761 | 0.789 | 0.769 | 0.751 | 0.578 | 0.544 |
| **HGDA** | **0.791** | **0.756** | **0.829** | **0.787** | **0.779** | **0.799** | **0.781** | **0.763** | **0.594** | **0.571** |

*Table 2.* Cross-network node classification on the Citation and Twitch network.

| Methods | US → CN | US → DE | US → JP | US → RU | US → FR | CN → US | CN → DE | CN → JP | CN → RU | CN → FR |
|---|---|---|---|---|---|---|---|---|---|---|
| GCN | 0.042 | 0.168 | 0.219 | 0.147 | 0.182 | 0.193 | 0.064 | 0.160 | 0.069 | 0.067 |
| DANN | 0.242 | 0.263 | 0.379 | 0.218 | 0.207 | 0.302 | 0.134 | 0.214 | 0.119 | 0.107 |
| DANE | 0.272 | 0.250 | 0.280 | 0.210 | 0.186 | 0.279 | 0.108 | 0.228 | 0.170 | 0.184 |
| UDAGCN | OOM | OOM | OOM | OOM | OOM | OOM | OOM | OOM | OOM | OOM |
| ASN | 0.290 | 0.272 | 0.291 | 0.222 | 0.199 | 0.268 | 0.121 | 0.207 | 0.189 | 0.190 |
| EGI | OOM | OOM | OOM | OOM | OOM | OOM | OOM | OOM | OOM | OOM |
| GRADE-N | 0.304 | 0.299 | 0.306 | 0.240 | 0.217 | 0.258 | 0.137 | 0.210 | 0.178 | 0.199 |
| JHGDA | OOM | OOM | OOM | OOM | OOM | OOM | OOM | OOM | OOM | OOM |
| SpecReg | 0.237 | 0.267 | 0.377 | 0.228 | 0.218 | 0.317 | 0.134 | 0.199 | 0.109 | 116 |
| PA | 0.400 | 0.389 | 0.474 | 0.371 | 0.252 | 0.452 | 0.262 | 0.383 | 0.333 | 0.242 |
| **HGDA**$_L$ | 0.463 | 0.442 | 0.498 | 0.400 | 0.314 | 0.447 | 0.379 | 0.418 | 0.368 | 0.330 |
| **HGDA**$_F$ | 0.449 | 0.457 | 0.502 | 0.416 | 0.329 | 0.441 | 0.401 | 0.395 | 0.388 | 0.307 |
| **HGDA**$_H$ | 0.488 | 0.468 | 0.494 | 0.429 | 0.330 | 0.426 | 0.410 | 0.409 | 0.326 | 0.288 |
| **HGDA** | **0.510** | **0.497** | **0.531** | **0.442** | **0.347** | **0.476** | **0.438** | **0.435** | **0.392** | **0.339** |

*Table 3.* Cross-network node classification on MAG datasets.

on Airport, 6.50% on ACM, 1.40% on Citation, and 2.20% on Blog. Notably, **HGDA** achieves a maximum average improvement of 8.49% for ACC on MAG datasets. This illustrates that our proposed model can effectively aligns varies node homophilic information. Our **HGDA** variants achieves second-highest than other optimal methods on some of the metrics in various benchmarks. This can be attributed to the varying homophilic pattern distributions across datasets, which are captured by our method's variants. For instance,

in the ACM dataset, **HGDA**$_L$ and **HGDA**$_H$ generally outperform **HGDA**$_F$, indicating that effectively balancing both homophilic and heterophilic information is crucial for this dataset. This observation aligns well with the design of our method. In all cases, **HGDA** produces better performance than PA (Liu et al., 2024c), which were published in 2024. This verifies the advantage of our approach. Our Model efficient experiment can be seen in Appendix D

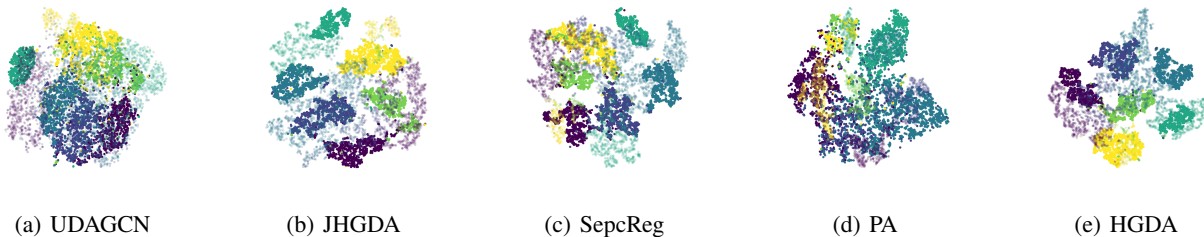

(a) UDAGCN      (b) JHGDA      (c) SepcReg      (d) PA      (e) HGDA

*Figure 4.* Visualizing source and target node embeddings via T-SNE. Each color represents a class, while dark and light shades of the same color correspond to nodes of the same class from the source and target domains, respectively.

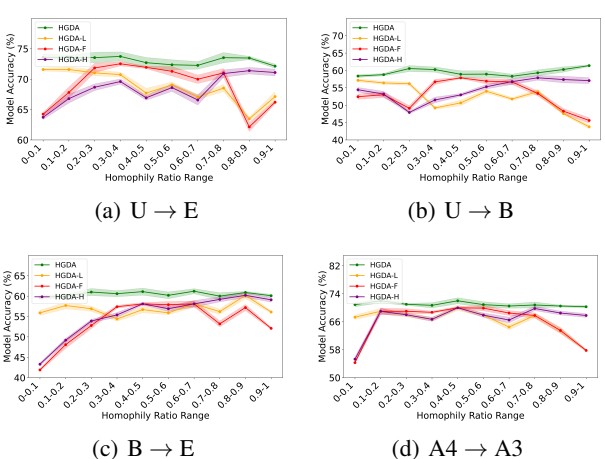

(a) U → E            (b) U → B

(c) B → E            (d) A4 → A3

*Figure 5.* Classification accuracy of **HGDA**, **HGDA**$_L$, **HGDA**$_F$, and **HGDA**$_L$ across different subgroups in Airport and ACM datasets.

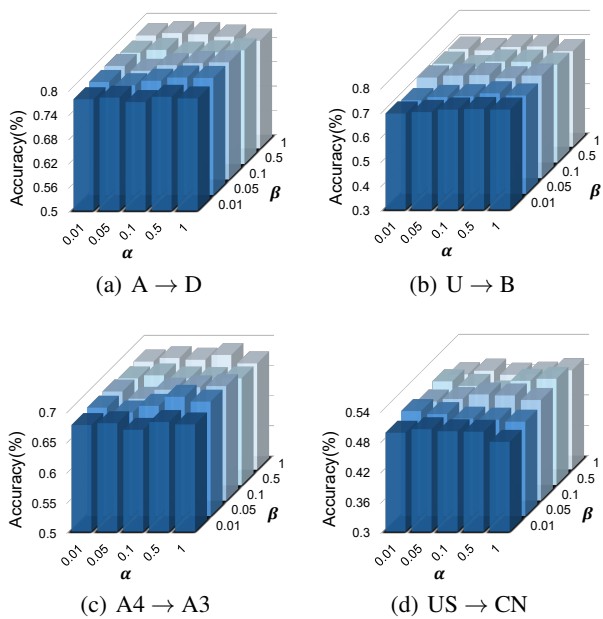

(a) A → D            (b) U → B

(c) A4 → A3         (d) US → CN

*Figure 6.* The influence of parameters $\alpha$ and $\beta$ on Citation, Airport, ACM and MAG datasets.

## 5.4. Ablation Study

Among the four variants of **HGDA**, **HGDA** performs the best in most cases. **HGDA**$_L$, **HGDA**$_F$, and **HGDA**$_H$ achieve varying performance levels across different groups of datasets, indicating that their effectiveness is closely related to the node homophilic distribution of the datasets. **HGDA** is consistently better than all variants, indicating each component boosts the performance. Moreover, we investigate all **HGDA** variant's classification accuracy across different homophilic subgroup. As shown in Figure 5, We observe that **HGDA** performs well across all subgroups, indicating that homophily alignment effectively addresses the shortcomings of GCN in handling various homophily discrepancies. In addition, **HGDA** variants effectively address homophily discrepancies at different levels. Specifically, **HGDA**$_L$ performs well in homophilic subgroups, while **HGDA**$_H$ excels in heterophilic subgroups, demonstrating the robustness of our model.

## 5.5. Visualization

In this section, we visualize node embeddings generated by competitive GDA models in the task A → D. We observe from Figure 4. Firstly, the nodes belonging to different classes are well-separated from each other. This shows that **HGDA** is effective in distinguishing between different classes in the embedding space. Secondly, nodes belonging to the same class from different domains are mostly overlapping, which indicates that HGDA could significantly reduce domain discrepancy. The first observation indicates good classification, while the latter suggests good domain alignment.

## 5.6. Parameter Analysis

In this section, we analyze the sensitivity of hyperparameters $\alpha$ and $\beta$ of our method on Citation, Airport, ACM, and MAG datasets. First, we test the performance with different $\alpha$ and $\beta$. As shown in Figure 6, **HGDA** has competitive performance on a large range of values, which suggests the stability of our method. For a more detailed

analysis and result, refer to the Appendix E.

## 6. Conclusion

In this paper, we propose HGDA framework to solve the GDA problem in cross-network node classification tasks. We reveal the importance of the homophily shift in GDA through both empirical and theoretical analysis. Our approach minimizes homophily distribution shifts by optimizing homophilic, heterophilic, and attribute signals. Comprehensive experiments verify the superiority of our approach. We will also delve deeper into graph domain adaptation theory to develop more powerful models by considering different architectures (Zhuo et al., 2023; 2024a; Yang et al., 2025; 2024b).

## Acknowledgments

This work is supported by the Natural Sciences and Engineering Research Council of Canada (NSERC) Discovery Grants program.

## Impact Statement

This paper presents work whose goal is to advance the field of Machine Learning. There are many potential societal consequences of our work, none which we feel must be specifically highlighted here.

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

## A. Proof

**Definition 4** (1-Wasserstein Distance of Node Heterophily Distributions). *Let $G = (\mathcal{V}, \mathcal{E}, A, X, Y)$ and $G' = (\mathcal{V}', \mathcal{E}', A', X', Y')$ be two graphs with node heterophily distributions:*

$$h_G \sim P_V, \quad h_{G'} \sim P_{V'}. \tag{15}$$

*We define their 1-Wasserstein Distance as:*

$$W_1(h_G, h_{G'}) = \inf_{\gamma \in \Pi(P_V, P_{V'})} \mathbb{E}_{(h, h') \sim \gamma}[|h - h'|], \tag{16}$$

*where:*

- $\Pi(P_V, P_{V'})$ *denotes the set of all joint distributions $\gamma(h, h')$ whose marginal distributions are $P_V$ and $P_{V'}$, respectively.*

- *This distance measures the minimum transportation cost required to transform the distribution $P_V$ into $P_{V'}$, where the transportation cost is given by $|h - h'|$.*

*This metric characterizes the optimal transport distance between the node heterophily distributions of two graphs and enables the comparison of their overall heterophily structures. Specifically:*

- *If $W_1(h_G, h_{G'}) \approx 0$, the two graphs have very similar heterophily distributions.*

- *If $W_1(h_G, h_{G'})$ is large, the heterophily structures of the two graphs are significantly different.*

**Corollary 2** (Wasserstein-1 Distance for Feature Distributions). *Let $P_S^F$ and $P_T^F$ be the empirical distributions of node features $f$ in the source graph $G^S$ and target graph $G^T$, respectively. The 1-Wasserstein distance between these distributions is given by:*

$$W_1(P_S^F, P_T^F) = \inf_{\gamma^F \in \Pi(P_S^F, P_T^F)} \mathbb{E}_{(i,j) \sim \gamma^F}[\|f_i - f_j\|], \tag{17}$$

*where $\Pi(P_S^F, P_T^F)$ is the set of all joint distributions $\gamma^F(i, j)$ with marginals $P_S^F$ and $P_T^F$. This measures the minimum transportation cost required to transform the feature distribution of $G^S$ into that of $G^T$.*

**Corollary 3** (Wasserstein-1 Distance for Node Heterophily Distributions). *Let $P_S^H$ and $P_T^H$ be the empirical distributions of node heterophily values $h$ in the source graph $G^S$ and target graph $G^T$, respectively. The 1-Wasserstein distance between these distributions is given by:*

$$W_1(P_S^H, P_T^H) = \inf_{\gamma^H \in \Pi(P_S^H, P_T^H)} \mathbb{E}_{(i,j) \sim \gamma^H}[|h_i - h_j|], \tag{18}$$

*where $\Pi(P_S^H, P_T^H)$ is the set of all joint distributions $\gamma^H(i, j)$ with marginals $P_S^H$ and $P_T^H$. This quantifies the minimum transportation cost to align the heterophily distribution between the two graphs.*

**Lemma 1** (Upper Bound on 1-Wasserstein Distance via KL Divergence). *(Bobkov & Götze, 1999) Let $P$ and $Q$ be two probability distributions on a metric space $(\mathcal{X}, d)$, where $P \ll Q$ (i.e., $P$ is absolutely continuous with respect to $Q$). Assume that:*

- $\mathcal{X}$ *has bounded support with diameter $D$, i.e., $d(x, y) \leq D$ for all $x, y \in \mathcal{X}$,*

- *There exists a constant $C_0 > 0$ such that all 1-Lipschitz functions $f : \mathcal{X} \to \mathbb{R}$ satisfy the concentration inequality:*

$$\sup_{f \in \mathcal{F}_L} \left| \mathbb{E}_{x \sim P}[f(x)] - \mathbb{E}_{x \sim Q}[f(x)] \right| \leq C_0 \sqrt{D_{KL}(P \| Q)}. \tag{19}$$

*Then, the 1-Wasserstein distance satisfies the following upper bound:*

$$W_1(P, Q) \leq C_0 \sqrt{D_{KL}(P \| Q)}. \tag{20}$$

*The constant $C$ depends on the geometry of the space $\mathcal{X}$ and the metric $d$.*

**Definition 5** (Expected Loss Discrepancy). *Given a distribution $P$ over a function family $\mathcal{H}$, for any $\lambda > 0$ and $\gamma \geq 0$, for any $G^S$ and $G^T$, define the expected loss discrepancy between $\mathcal{V}^S$ and $\mathcal{V}^T$ as $D_{S,T}^\gamma(P;\lambda) := \ln \mathbb{E}_{\phi \sim P} e^{\lambda \left( \mathcal{L}_T^{\gamma/2}(\phi) - \mathcal{L}_S^\gamma(\phi) \right)}$, where $\mathcal{L}_T^{\gamma/2}(\phi)$ and $\mathcal{L}_S^\gamma(\phi)$ follow the definition of Eq. (3).*

**Lemma 2** (Adaptation from (Ma et al., 2021)). *Let $\Phi$ be a family of classification functions. For any classifier $\phi$ in $\Phi$, and for any parameters $\lambda > 0$ and $\gamma \geq 0$, consider any prior distribution $P$ over $\Phi$ that is independent of the training data $\mathcal{V}^S$. With a probability of at least $1 - \delta$ over the sample $Y^S$, for any distribution $Q$ on $\Phi$ such that $\Pr_{\phi \sim Q} \left[ \max_{i \in \mathcal{V}^S \cup \mathcal{V}^T} \|\phi_i(X,G) - \phi_i(X,G)\|_\infty < \frac{\gamma}{8} \right] > \frac{1}{2}$, the following inequality holds:*

$$\mathcal{L}_T^0(\phi) \leq \widehat{\mathcal{L}}_S^\gamma(\phi) + \frac{1}{\lambda} \left[ 2(D_{\mathrm{KL}}(Q\|P) + 1) + \ln \frac{1}{\delta} + \frac{\lambda^2}{4N_S} + D_{S,T}^{\gamma/2}(P;\lambda) \right].$$

**Proposition 1** (Bound for $D_{S,T}^\gamma(P;\lambda)$, Adaptation from (Ma et al., 2021; Mao et al., 2024a; Fang et al., 2025)). *For any $\gamma \geq 0$, and under the assumption that the prior distribution $P$ over the classification function family $\Phi$ is defined, we establish a bound for the domain discrepancy measure $D_{S,T}^{\gamma/2}(P;\lambda)$. Specifically, we have the following inequality:*

$$
\begin{aligned}
D_{S,T}^{\gamma/2}(P;\lambda) &\leq \frac{1}{\max(N_S, N_T)} \sum_{i \in V_S} \frac{1}{N_T} \sum_{j \in V_T} \left[ \ln 3 + \frac{\lambda \rho C}{\sqrt{2\pi}\sigma}(\|f_i - f_j\| + |h_i - h_j| \cdot \rho) \right] \\
&\leq \frac{\lambda \rho C}{\sqrt{2\pi}\sigma \cdot N_S \cdot N_T} \cdot W_1(P_S^F, P_T^F) + W_1(P_S^H, P_T^H) \\
&\leq \frac{\lambda \rho C C_0}{\sqrt{2\pi}\sigma \cdot N_S \cdot N_T} \cdot \left[ \sqrt{D_{KL}(P_S^F\|P_T^F)} + \sqrt{D_{KL}(P_S^H\|P_T^H)} \right]
\end{aligned}
\tag{21}
$$

*where $f$ is the unspecified operator of first-order aggregation ( considering $f = AX$ and $f = LX$ two easy cases), $C$ is the number of classes, and $\rho, \sigma$ are constants controlled by the node feature distribution of different classes.*

*Proof.* We will prove the proposition by establishing the two key inequalities in the given equation.

**Proof of the First Inequality (Using Prior Works):**
From the cited works (Ma et al., 2021; Mao et al., 2024a; Fang et al., 2025), it follows that the expected loss discrepancy measure $D_{S,T}^{\gamma/2}(P;\lambda)$ can be bounded by:

$$D_{S,T}^{\gamma/2}(P;\lambda) \leq \frac{1}{\max(N_S, N_T)} \sum_{i \in V_S} \frac{1}{N_T} \sum_{j \in V_T} \left[ \ln 3 + \frac{\lambda C \rho}{\sqrt{2\pi}\sigma}(\|f_i - f_j\| + |h_i - h_j| \cdot \rho) \right]. \tag{22}$$

This follows from existing domain adaptation literature, where the domain discrepancy measure can be controlled by a sum over pairwise differences of node features and heterophily values.

**Proof of the Second Inequality:**
We now show that:

$$\frac{1}{\max(N_S, N_T)} \sum_{i \in V_S} \frac{1}{N_T} \sum_{j \in V_T} \left[ \ln 3 + \frac{\lambda C \rho}{\sqrt{2\pi}\sigma}(\|f_i - f_j\| + |h_i - h_j| \cdot \rho) \right] \tag{23}$$

$$\leq \frac{\lambda C \rho}{\sqrt{2\pi}\sigma \cdot N_S \cdot N_T} \left( W_1(P_S^F, P_T^F) + \rho W_1(P_S^H, P_T^H) \right). \tag{24}$$

From **Corollary 2**, we know that the Wasserstein-1 distance between the node feature distributions is:

$$W_1(P_S^F, P_T^F) = \inf_{\gamma^F \in \Pi(P_S^F, P_T^F)} \mathbb{E}_{(i,j) \sim \gamma^F}[\|f_i - f_j\|]. \tag{25}$$

Since the sum over all node pairs approximates an expectation over a joint coupling of the distributions, we approximate:

$$\sum_{i \in V_S} \sum_{j \in V_T} \|f_i - f_j\| \approx N_S N_T W_1(P_S^F, P_T^F). \tag{26}$$

Substituting this into our sum, we obtain:

$$\sum_{i\in V_S}\sum_{j\in V_T}\frac{1}{N_T}\|f_i - f_j\| \le N_S W_1(P_S^F, P_T^F). \tag{27}$$

Similarly, from **Corollary 3**, we have:

$$W_1(P_S^H, P_T^H) = \inf_{\gamma^H\in\Pi(P_S^H,P_T^H)}\mathbb{E}_{(i,j)\sim\gamma^H}[|h_i - h_j|]. \tag{28}$$

Following the same argument as before, we approximate:

$$\sum_{i\in V_S}\sum_{j\in V_T}|h_i - h_j| \approx N_S N_T W_1(P_S^H, P_T^H). \tag{29}$$

Thus:

$$\sum_{i\in V_S}\sum_{j\in V_T}\frac{1}{N_T}|h_i - h_j| \le N_S W_1(P_S^H, P_T^H). \tag{30}$$

Now substituting these bounds into our sum:

$$\sum_{i\in V_S}\sum_{j\in V_T}\left[\ln 3 + \frac{\lambda C\rho}{\sqrt{2\pi}\sigma}(\|f_i - f_j\| + |h_i - h_j|\cdot\rho)\right] \tag{31}$$

$$\le \sum_{i\in V_S}\sum_{j\in V_T}\ln 3 + \frac{\lambda C\rho}{\sqrt{2\pi}\sigma}N_S(W_1(P_S^F, P_T^F) + \rho W_1(P_S^H, P_T^H)). \tag{32}$$

Dividing by $\max(N_S, N_T)$, we obtain:

$$\frac{1}{\max(N_S, N_T)}\sum_{i\in V_S}\sum_{j\in V_T}\left[\ln 3 + \frac{\lambda C\rho}{\sqrt{2\pi}\sigma}(\|f_i - f_j\| + |h_i - h_j|\cdot\rho)\right] \tag{33}$$

$$\le \frac{\lambda C\rho}{\sqrt{2\pi}\sigma\cdot N_S\cdot N_T}(W_1(P_S^F, P_T^F) + \rho W_1(P_S^H, P_T^H)). \tag{34}$$

**Proof of the Third Inequality:**
From **Lemma 1** (Bobkov-Götze Inequality) (Bobkov & Götze, 1999), we have:

$$W_1(P_S^F, P_T^F) \le C_0\sqrt{D_{\mathrm{KL}}(P_S^F\|P_T^F)}, \tag{35}$$

$$W_1(P_S^H, P_T^H) \le C_0\sqrt{D_{\mathrm{KL}}(P_S^H\|P_T^H)}. \tag{36}$$

Using these bounds in:

$$\frac{\lambda\rho C}{\sqrt{2\pi}\sigma\cdot N_S\cdot N_T}\cdot W_1(P_S^F, P_T^F) + \rho W_1(P_S^H, P_T^H), \tag{37}$$

we obtain:

$$\frac{\lambda\rho CC_0}{\sqrt{2\pi}\sigma\cdot N_S\cdot N_T}\cdot\left[\sqrt{D_{\mathrm{KL}}(P_S^F\|P_T^F)} + \sqrt{D_{\mathrm{KL}}(P_S^H\|P_T^H)}\right]. \tag{38}$$

Combining both inequalities, we establish the bound:

$$D_{S,T}^{\gamma/2}(P;\lambda) \le \frac{\lambda\rho CC_0}{\sqrt{2\pi}\sigma\cdot N_S\cdot N_T}\cdot\left[\sqrt{D_{\mathrm{KL}}(P_S^F\|P_T^F)} + \sqrt{D_{\mathrm{KL}}(P_S^H\|P_T^H)}\right] \tag{39}$$

This completes the proof. □ □

**Corollary 4** (KL Divergence Decomposition of Feature Distributions). *Let $P_S^F$ and $P_T^F$ be the feature distributions in the source and target domains, respectively. Suppose that node features $X$ undergo transformations defined by the adjacency matrix $A$ and Laplacian matrix $L$, i.e.,*

$$P_S^F = P(A^S X^S, L^S X^S | X^S) P(X^S), \quad P_T^F = P(A^T X^T, L^T X^T | X^T) P(X^T). \tag{40}$$

*Then, the KL divergence between the feature distributions satisfies:*

$$D_{KL}(P_S^F \| P_T^F) \le D_{KL}(A^S X^S \| A^T X^T) + D_{KL}(X^S \| X^T) + D_{KL}(L^S X^S \| L^T X^T). \tag{41}$$

*Proof.* Using the chain rule for KL divergence, we decompose:

$$D_{\mathrm{KL}}(P_S^F \| P_T^F) = D_{\mathrm{KL}}(P(A^S X^S, L^S X^S | X^S) P(X^S) \| P(A^T X^T, L^T X^T | X^T) P(X^T)). \tag{42}$$

Applying the chain rule:

$$D_{\mathrm{KL}}(P_S^F \| P_T^F) = D_{\mathrm{KL}}(P(X^S) \| P(X^T)) + \mathbb{E}_{P(X^S)} D_{\mathrm{KL}}(P(A^S X^S, L^S X^S | X^S) \| P(A^T X^T, L^T X^T | X^T)). \tag{43}$$

Assuming conditional independence between adjacency and Laplacian transformations:

$$D_{\mathrm{KL}}(P(A^S X^S, L^S X^S | X^S) \| P(A^T X^T, L^T X^T | X^T)) \le D_{\mathrm{KL}}(A^S X^S \| A^T X^T) + D_{\mathrm{KL}}(L^S X^S \| L^T X^T). \tag{44}$$

Thus, combining the results:

$$D_{\mathrm{KL}}(P_S^F \| P_T^F) \le D_{\mathrm{KL}}(A^S X^S \| A^T X^T) + D_{\mathrm{KL}}(X^S \| X^T) + D_{\mathrm{KL}}(L^S X^S \| L^T X^T). \tag{45}$$

This completes the proof. $\qquad\qquad\qquad\qquad\qquad\qquad\qquad\square$ $\qquad\qquad\qquad\qquad\square$

**Lemma 3.** *[Domain Adaptation Bound for Deterministic Classifiers] Let $\Phi$ be a family of classification functions. For any classifier $\phi$ in $\Phi$, and for any parameters $\lambda > 0$ and $\gamma \ge 0$, consider any prior distribution $P$ over $\Phi$ that is independent of the training data $\mathcal{V}^S$. With a probability of at least $1 - \delta$ over the sample $Y^S$, for any distribution $Q$ on $\Phi$ such that the following inequality holds:*

$$\mathcal{L}_T^0(\phi) \le \widehat{\mathcal{L}}_S^\gamma(\phi) + \frac{1}{\lambda} \left[ 2(D_{\mathrm{KL}}(Q\|P) + 1) + \ln\frac{1}{\delta} \right] \tag{46}$$

$$+ \frac{\lambda}{4N_S} + D_{S,T}^{\gamma/2}(P; \lambda) \tag{47}$$

**Theorem 5.** *Let $\Phi$ be a family of classification functions. For any classifier $\phi \in \Phi$, and for any parameters $\lambda > 0$ and $\gamma \ge 0$, consider any prior distribution $P$ over $\Phi$ that is independent of the training data $\mathcal{V}^S$. With probability at least $1 - \delta$ over the sample $Y^S$, for any distribution $Q$ on $\Phi$, we have:*

$$\mathcal{L}_T^0(\phi) \le \widehat{\mathcal{L}}_S^\gamma(\phi) + \frac{1}{\lambda} \left[ 2(D_{\mathrm{KL}}(Q\|P) + 1) + \ln\frac{1}{\delta} \right] + \frac{\lambda}{4N_S} + \frac{\lambda \rho C C_0}{\sqrt{2\pi}\sigma \cdot N_S \cdot N_T} \cdot$$
$$\left[ \sqrt{D_{KL}(A^S X^S \| A^T X^T)} + \sqrt{D_{KL}(X^S \| X^T)} + \sqrt{D_{KL}(L^S X^S \| L^T X^T)} + \sqrt{D_{KL}(P_S^H \| P_T^H)} \right]. \tag{48}$$

*Proof.* By Lemma 3, the domain adaptation bound is given by:

$$\mathcal{L}_T^0(\phi) \le \widehat{\mathcal{L}}_S^\gamma(\phi) + \frac{1}{\lambda} \left[ 2(D_{\mathrm{KL}}(Q\|P) + 1) + \ln\frac{1}{\delta} \right] + \frac{\lambda}{4N_S} + D_{S,T}^{\gamma/2}(P; \lambda). \tag{49}$$

From Proposition 1, we substitute the bound on $D_{S,T}^{\gamma/2}(P; \lambda)$:

$$D_{S,T}^{\gamma/2}(P; \lambda) \le \frac{\lambda \rho C C_0}{\sqrt{2\pi}\sigma \cdot N_S \cdot N_T} \left[ \sqrt{D_{\mathrm{KL}}(P_S^F \| P_T^F)} + \sqrt{D_{\mathrm{KL}}(P_S^H \| P_T^H)} \right]. \tag{50}$$

Applying Corollary 4 to decompose $D_{\mathrm{KL}}(P_S^F \| P_T^F)$, we obtain:

$$D_{\mathrm{KL}}(P_S^F \| P_T^F) \leq D_{\mathrm{KL}}(A^S X^S \| A^T X^T) + D_{\mathrm{KL}}(X^S \| X^T) + D_{\mathrm{KL}}(L^S X^S \| L^T X^T). \tag{51}$$

Similarly, for the heterophily-based KL divergence:

$$D_{\mathrm{KL}}(P_S^H \| P_T^H) \leq D_{\mathrm{KL}}(A^S X^S \| A^T X^T) + D_{\mathrm{KL}}(X^S \| X^T) + D_{\mathrm{KL}}(L^S X^S \| L^T X^T). \tag{52}$$

Thus, we obtain:

$$D_{S,T}^{\gamma/2}(P; \lambda) \leq \frac{\lambda \rho C C_0}{\sqrt{2\pi}\sigma \cdot N_S \cdot N_T} \left[ \sqrt{D_{\mathrm{KL}}(A^S X^S \| A^T X^T)} + \sqrt{D_{\mathrm{KL}}(X^S \| X^T)} + \sqrt{D_{\mathrm{KL}}(L^S X^S \| L^T X^T)} \right]. \tag{53}$$

Substituting this into the original bound for $\mathcal{L}_T^0(\phi)$, we conclude:

$$\begin{aligned}
\mathcal{L}_T^0(\phi) \leq \widehat{\mathcal{L}}_S^\gamma(\phi) + \frac{1}{\lambda}\left[ 2(D_{\mathrm{KL}}(Q\|P) + 1) + \ln\frac{1}{\delta} \right] + \frac{\lambda}{4N_S} \\
+ \frac{\lambda \rho C C_0}{\sqrt{2\pi}\sigma \cdot N_S \cdot N_T} \left[ \sqrt{D_{\mathrm{KL}}(A^S X^S \| A^T X^T)} + \sqrt{D_{\mathrm{KL}}(X^S \| X^T)} + \sqrt{D_{\mathrm{KL}}(L^S X^S \| L^T X^T)} \right].
\end{aligned} \tag{54}$$

This completes the proof. $\square$ $\square$

## B. Experimental Setup

The experiments are implemented in the PyTorch platform using an Intel(R) Xeon(R) Silver 4210R CPU @ 2.40GHz, and GeForce RTX A5000 24G GPU. Technically, two layers GCN is built and we train our model by utilizing the Adam (Kingma & Ba, 2015) optimizer with learning rate ranging from 0.0001 to 0.0005. In order to prevent over-fitting, we set the dropout rate to 0.5. In addition, we set weight decay $\in \{1e-4, \cdots, 5e-3\}$. For fairness, we use the same parameter settings for all the cross-domain node classification methods in our experiment, except for some special cases. For GCN, UDA-GCN, and JHGDA the GCNs of both the source and target networks contain two hidden layers ($L=2$) with structure as $128-16$. The dropout rate for each GCN layer is set to $0.3$. We repeatedly train and test our model for five times with the same partition of dataset and then report the average of ACC.

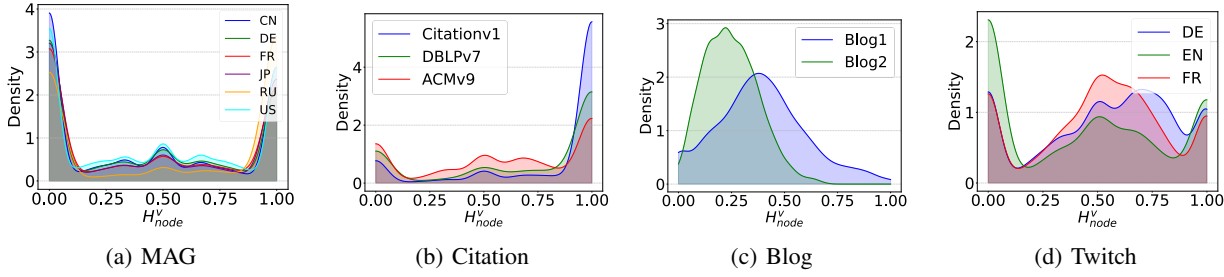

*Figure 7.* This shows the local node homophily distibution shift is existing in MAG, Citation, Blog and Twitch datasets.

## C. Empirical Study: Homophily Divergence Effect to Target classification Performance

In this appendix, we extend our analysis by presenting the homophily distributions and performance results for additional datasets, including Blog, Citation, Twitch, and MAG. As shown in Figure 7, these datasets exhibit clear shifts in homophily distributions between the source and target graphs, further underlining the necessity of our proposed GDA method. The experiments focus on assessing the impact of homophily divergence on target node classification performance. Using a standard two-layer GCN with unsupervised GDA settings, we evaluated classification accuracy across varying homophily ratio subgroups. The following detail observations are drawn from the analysis:

**Homophily Distribution Differences:** Similar to the Airport dataset discussed in the main text, the Blog, Citation, Twitch, and MAG datasets show significant divergence in homophily ratios between the source and target graphs. These shifts result in node subgroups with varying levels of homophilic and heterophilic groups, as illustrated in Figure 7.

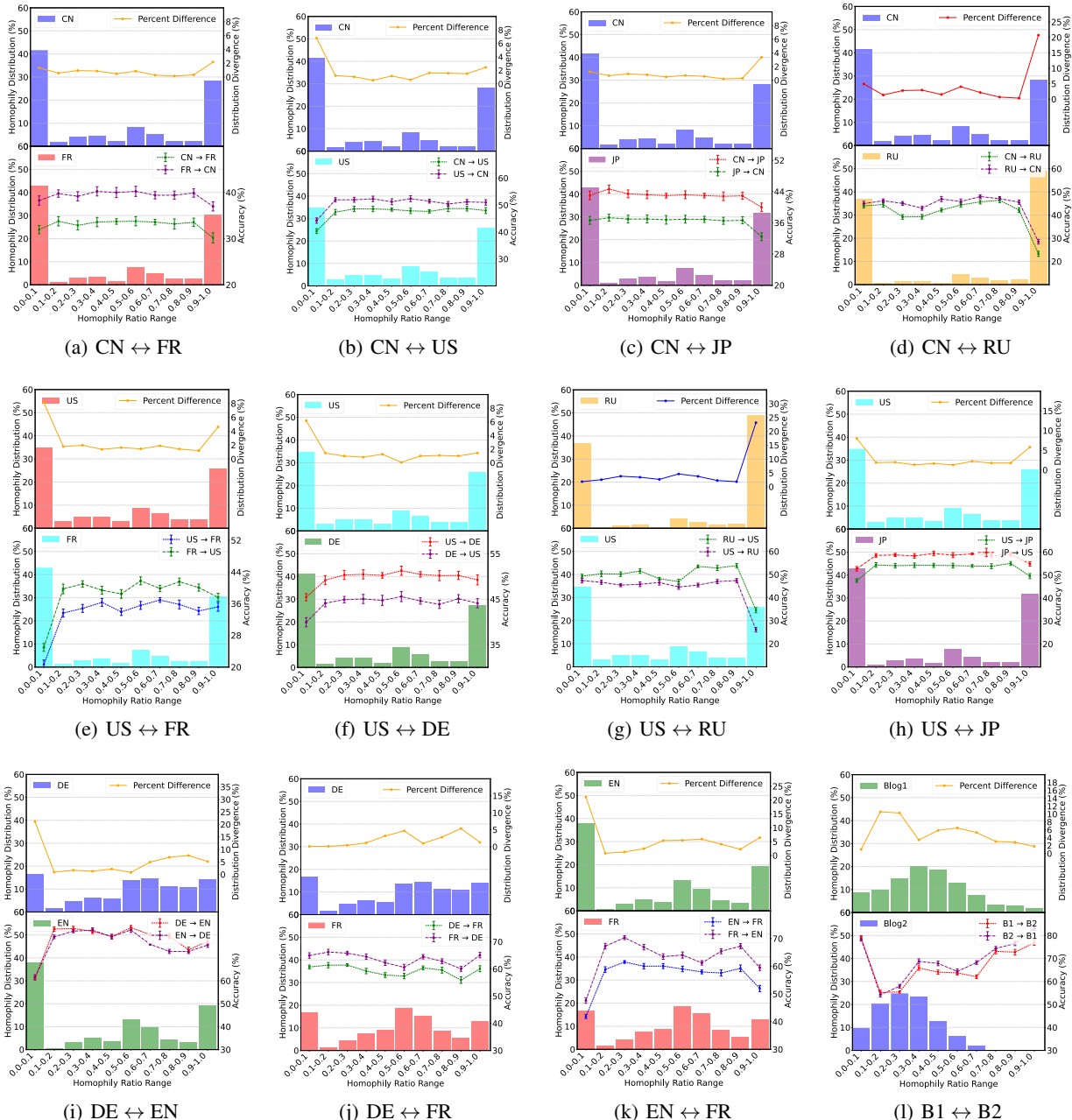

*Figure 8.* GCN performance on cross-network classification tasks across different homophily ratio ranges for the MAG, Twitch, and Blog datasets.

| Dataset | Method | Training Time (Normalized w.r.t. UDAGCN) | Memory Usage (Normalized w.r.t. UDAGCN) | Accuracy(%) |
|---|---|---|---|---|
| U→B | UDAGCN | 1 | 1 | 0.607 |
| | JHGDA | 1.314 | 1.414 | 0.695 |
| | PA | 0.498 | 0.517 | 0.481 |
| | SpecReg | 0.463 | 0.493 | 0.621 |
| | *HGDA* | **0.514** | **0.314** | **0.721** |
| U→E | UDAGCN | 1 | 1 | 0.472 |
| | JHGDA | 1.423 | 1.513 | 0.519 |
| | PA | 0.511 | 0.509 | 0.547 |
| | SpecReg | **0.517** | 0.503 | 0.487 |
| | *HGDA* | **0.307** | **0.310** | **0.572** |
| B→E | UDAGCN | 1 | 1 | 0.497 |
| | JHGDA | 1.311 | 1.501 | 0.569 |
| | PA | 0.502 | 0.497 | 0.516 |
| | SpecReg | 0.407 | 0.503 | 0.546 |
| | *HGDA* | **0.309** | **0.313** | **0.584** |
| A→D | UDAGCN | 1 | 1 | 0.510 |
| | JHGDA | 1.311 | 1.501 | 0.569 |
| | PA | 0.502 | 0.497 | 0.562 |
| | SpecReg | 0.407 | 0.503 | 0.536 |
| | *HGDA* | **0.309** | **0.313** | **0.791** |
| A→C | UDAGCN | 1 | 1 | 0.510 |
| | JHGDA | 1.311 | 1.501 | 0.569 |
| | PA | 0.502 | 0.497 | 0.562 |
| | SpecReg | 0.407 | 0.503 | 0.536 |
| | *HGDA* | **0.309** | **0.313** | **0.326** |
| C→D | UDAGCN | 1 | 1 | 0.510 |
| | JHGDA | 1.311 | 1.501 | 0.570 |
| | PA | 0.502 | 0.497 | 0.562 |
| | SpecReg | 0.407 | 0.503 | 0.536 |
| | *HGDA* | **0.309** | **0.313** | **0.326** |

*Table 4.* Comparison of Training Time, Memory Usage, and Accuracy on Airport datset.

**Impact on Classification Accuracy:** we use a two-layer GCN with standard unsupervised GDA settings as our evaluation model. After completing model training, we report the classification accuracy for different homophily ratio subgroups. In the figure, the x-axis represents different node subgroups, where each subgroup consists of nodes within specified homophily ratios range. The left y-axis denotes the proportion of nodes in the entire graph, while the upper-right y-axis shows the percentage difference in homophily ratios between the source and target graphs for each subgroup. The lower-right y-axis indicates the GDA node classification accuracy on the target graph for each subgroup. As shown in Figure 8, classification accuracy on the target graph depends on the proportion difference of nodes belonging to subgroups with varying homophily ratios. The figure illustrates that subgroups with lower homophily ratio differences achieve better classification performance, while those that deviate more from the source graph tend to exhibit lower accuracy. Experimental results reveal a negative correlation between homophily divergence and GDA performance. More significant disparities in homophily ratios between the source and target graphs lead to lower classification accuracy on the target graph. This trend underscores the importance of GDA in accounting for homophily discrepancies across domains.

# D. Model Efficient Experiment

**Model Complexity**: Here, we analyze the computational complexity of the proposed HGDA. The computational complexity primarily depends on the three filter layers. For a given graph $G$, let $N$ represent the total number of nodes in the graph and $d$ the feature dimension. For the three filters: Homophilic Filter $H_L$: Includes matrix multiplication with $W_L^{l-1}$ ($O(N^2 \cdot d)$) . Heterophilic Filter $H_H$: $O(N^2 \cdot d)$. Full-pass Filter $H_F$: $O(N^2 \cdot d)$. Thus, the total complexity of HGDA is: $O(N^2 \cdot d)$. Since HGDA does not introduce complex modules, its model complexity remains low, comparable to that of MLP or GCN.

**Model Efficient Experiment**: To further evaluate the efficiency of HGDA, Table 4 presents a running time comparison across various algorithms. We also compared the GPU memory usage of common baselines, including UDAGCN and the recent state-of-the-art methods JHGDA, PA, and SpecReg, which align graph domain discrepancies through different ways. As shown in Table, the evaluation results on airport and citation dataset further demonstrate that our method achieves superior performance with tolerable computational and storage overhead.

# E. Parameter Analysis

The effectiveness of Graph Domain Adaptation (GDA) is highly influenced by the homophily distribution differences between the source and target graphs. Homophily ratios, defined as the likelihood of connected nodes sharing similar attributes or labels, play a crucial role in determining how well embeddings align across domains. To address these variations, the hyperparameters $\alpha$ and $\beta$ are selected from the set $\{0.01, 0.05, 0.1, 0.5, 1\}$, providing flexibility in balancing the contributions of source label supervision, domain alignment, and target graph adaptation.

**Airport Dataset:** The Airport dataset represents transportation networks, typically characterized by a smaller number of nodes with complex edge relationships and sparse connectivity. The homophily ratio in this dataset is relatively low, which highlights the importance of capturing key node alignment information and homophilic relationships to maintain meaningful structure. The sparsity of connections and low homophily ratios require higher $\alpha$ and $\beta$ values to emphasize alignment between source and target graphs. This ensures that even sparse but critical homophilic relationships are preserved. $\alpha$ and $\beta$ are chosen from $\{0.05, 0.1, 0.5\}$ to prioritize node alignment and structural consistency.

**Citation Dataset (Citation and ACM Datasets):** The Citation dataset often exhibits diverse structural homophily patterns across different distributions, with varying levels of similarity between source and target graphs. These variations necessitate a careful balance between domain alignment and label information from the source graph. The diversity in homophily ratios between the source and target graphs requires moderate $\alpha$ and $\beta$ values to balance domain alignment loss ($\mathcal{L}_H$) and source classifier loss ($\mathcal{L}_S$). Misalignment in homophily patterns can negatively impact the knowledge transfer process if not adequately addressed. $\alpha$ and $\beta$ are selected from $\{0.1, 0.5\}$, ensuring an effective trade-off between domain alignment and label supervision.

**Social Network Dataset (Blog and Twitch Datasets):** Social networks are characterized by a large number of nodes with rich attribute information and high variability in structural patterns. These datasets often feature distinct attribute distributions, making attribute alignment crucial for successful domain adaptation.Social networks typically exhibit higher variance in homophily ratios across source and target domains. This makes alignment particularly sensitive to changes in $\alpha$ and $\beta$. Lower values are recommended to prevent overfitting to either homophily and heterophilic patterns.$\alpha$ and $\beta$ are chosen from $\{0.05, 0.1\}$ to focus on attribute alignment while maintaining flexibility for homophily shifts.

**MAG Dataset:** The MAG dataset is large and diverse, containing numerous classes with intricate relationships and rich metadata. Both homophily and heterophily alignment play critical roles in ensuring effective knowledge transfer across domains. Due to its diverse nature, the MAG dataset exhibits a mix of homophily and heterophily patterns. Alignment must consider both structural consistency and attribute-based adaptation to account for this diversity. Misalignment in either homophily or heterophily can lead to suboptimal performance. $\alpha$ and $\beta$ are selected from $\{0.1, 0.5\}$ to balance the importance of both homophily and heterophily alignment, ensuring robust performance across diverse classes and relationships.

# F. Description of Algorithm HGDA

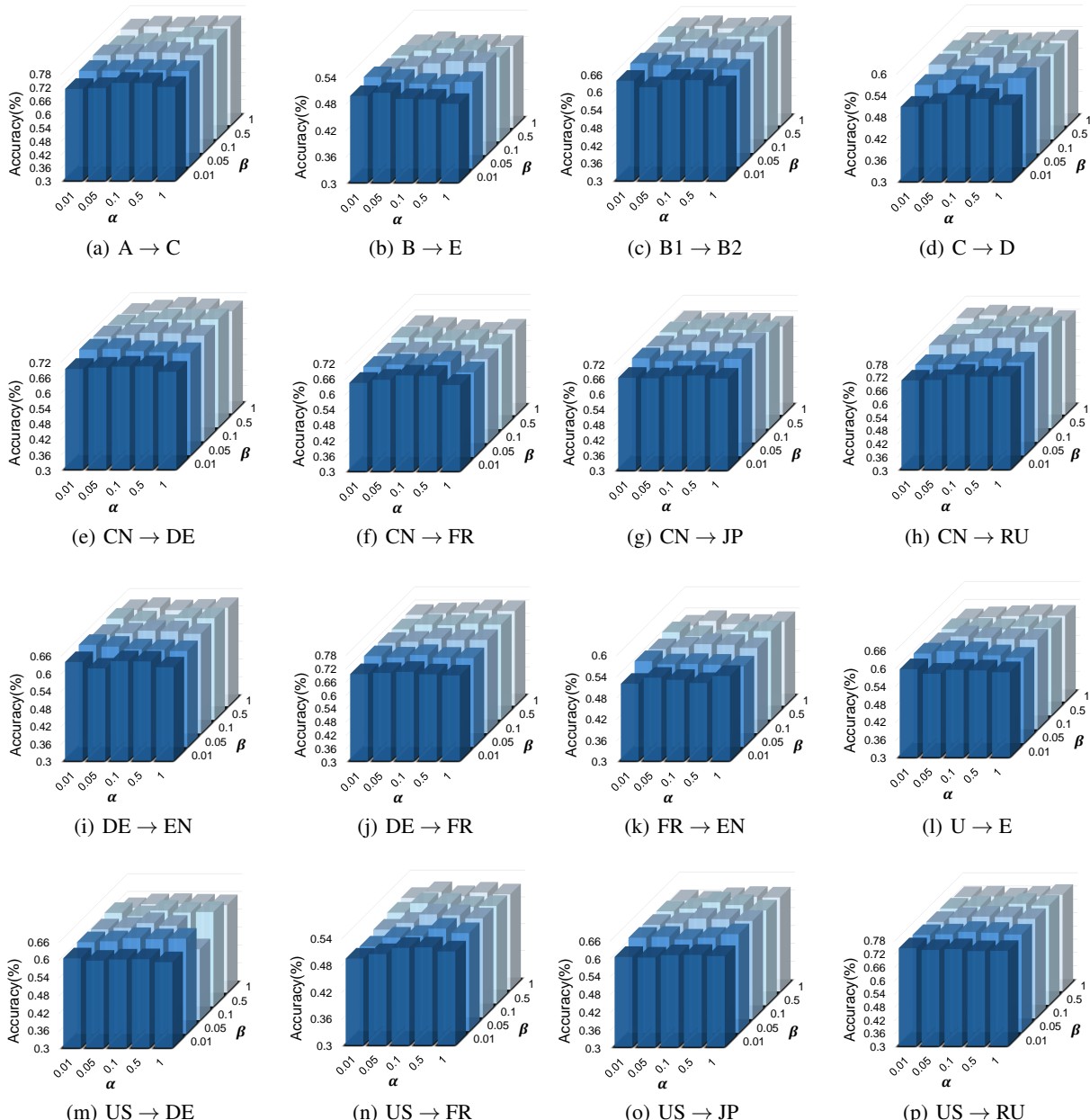

*Figure 9.* The influence of hyper-parameters $\alpha$ and $\beta$ on Citation, Airport, ACM and MAG datasets.

---

**Algorithm 1** The proposed algorithm HGDA

---

**Input:** Source node feature matrix $X^S$; source original graph adjacency matrix $A^S$; Target node feature matrix $X^T$; Target original graph adjacency matrix $A^T$ source node label matrix $Y^S$; maximum number of iterations $\eta$

1  Compute the $\tilde{A}$ and $\tilde{L}$ according to $G^S$ and $G^T$ by running .

2  **for** $it = 1$ **to** $\eta$ **do**

3      $Z^S_L = H_L(\tilde{A}^S, X^S)$

4      $Z^S_F = H_F(I^S, X^S)$

5      $Z^S_H = H_H(\tilde{L}^S, X^S)$
       `// embedding of source graph`

6      $Z^T_L = H_L(\tilde{A}^T, X^T)$

7      $Z^T_F = H_F(I^T, X^T)$

8      $Z^T_H = H_H(\tilde{L}^T, X^T)$
       `// embedding of target graph`

9      Homophily enhanced domain adaptive between $Z^S_L$ and $Z^T_L$, $Z^S_F$ and $Z^T_F$ and $Z^S_H$ and $Z^T_H$
       `// adaptive in three views`

10     Source graph classification $Z^S$ and $Y^S$

11     Domain Adaptive Learning between $Z^S$ and $Z^T$

12     Calculate the overall loss with Eq.(14)

13     Update all parameters of the framework according to the overall loss

14 Predict the labels of target graph nodes based on the trained framework.

**Output:** Classification result $\hat{Y}^T$

---

| Types | Datasets | $\alpha$ | $\beta$ |
|---|---|---|---|
| Airport | U→B | 0.5 | 0.5 |
| | U→E | 0.05 | 0.1 |
| | B→U | 0.1 | 0.1 |
| | B→E | 0.5 | 0.1 |
| | E→U | 0.5 | 0.5 |
| | E→B | 0.5 | 0.5 |
| Citation | A→D | 0.1 | 0.1 |
| | D→A | 0.1 | 0.1 |
| | A→C | 0.5 | 0.5 |
| | C→A | 0.1 | 0.1 |
| | C→D | 0.1 | 0.1 |
| | D→C | 0.1 | 0.1 |
| Blog | B1→B2 | 0.1 | 0.1 |
| | B2→B1 | 0.1 | 0.1 |
| Twitch | DE→EN | 0.1 | 0.1 |
| | EN→DE | 0.1 | 0.5 |
| | DE→FR | 0.5 | 0.5 |
| | FR→EN | 0.1 | 0.1 |
| Blog | B1→B2 | 0.1 | 0.1 |
| | B2→B1 | 0.1 | 0.1 |
| MAG | US→CN | 0.5 | 0.1 |
| | US→DE | 0.1 | 0.1 |
| | US→JP | 0.1 | 0.5 |
| | US→RU | 0.1 | 0.1 |
| | US→FR | 0.1 | 0.1 |
| | CN→US | 0.5 | 0.1 |
| | CN→DE | 0.1 | 0.1 |
| | CN→JP | 0.1 | 0.1 |
| | CN→RU | 0.5 | 0.1 |
| | CN→FR | 0.1 | 0.01 |

*Table 5.* Experiment hyperparameter setting Value.

| Types | Datasets | #Node | #Edge | #Label |
|-------|----------|-------|-------|--------|
| Airport | USA | 1,190 | 13,599 | |
| | Brazil | 131 | 1,038 | 4 |
| | Europe | 399 | 5,995 | |
| Citation | ACMv9 | 9,360 | 15,556 | |
| | Citationv1 | 8,935 | 15,098 | 5 |
| | DBLPv7 | 5,484 | 8,117 | |
| Blog | Blog1 | 2,300 | 33,471 | 6 |
| | Blog2 | 2,896 | 53,836 | |
| Twitter | Germany | 9,498 | 153,138 | |
| | England | 7,126 | 35,324 | 2 |
| | France | 6,549 | 112,666 | |
| ACM | ACM3 | 3,025 | 2,221,699 | 3 |
| | ACM4 | 4019 | 57,853 | |
| MAG | US | 132,558 | 697,450 | |
| | CN | 101,952 | 285,561 | |
| | DE | 43,032 | 126,683 | 20 |
| | JP | 37,498 | 90,944 | |
| | RU | 32,833 | 67,994 | |
| | FR | 29,262 | 78,222 | |

*Table 6.* Dataset Statistics.

