# OpenReview forum: "Homophily Enhanced Graph Domain Adaptation"
_ICML.cc/2025/Conference — ICML 2025 poster_

### Official Review · Reviewer_oEYM · 2025-03-06

**Overall Recommendation:** 3

**Summary:**

This paper investigates graph domain adaptation through homophily alignment. The authors argued that the graph homophily has been overlooked by existing graph domain adaptation works. To address this issue, the authors first conduct some empirical analyses and find that the homophily discrepancies indeed exist in the widely used benchmarks. Then, the authors proposed a model to use mixed filters to smooth graph signals and align homophily discrepancies between source and target graphs. Experimental results on 5 public datasets show that it can achieve satisfied performance compared with recent baselines.

**Claims And Evidence:**

The key claim is not convincing:

The authors propose to utilize three filters to learn graph signals, which is defined as homophilic filter, full-pass filter and heterophilic filter. Based on the Equations (8), (9) and (10), the authors use $AX$, $IX$ and $LX$ to represent the homophilic, full and heterophilic signals in graph. Why Laplacian matrix $L$ can represent the heterophilic signals, which is contradictive to the graph spectral theory.

**Essential References Not Discussed:**

The authors fail to compare and discuss the following highly relevant papers:

[1] Huang R, Xu J, Jiang X, et al. Can Modifying Data Address Graph Domain Adaptation?[C]//Proceedings of the 30th ACM SIGKDD Conference on Knowledge Discovery and Data Mining. 2024: 1131-1142.

[2] Chen W, Ye G, Wang Y, et al. Smoothness Really Matters: A Simple yet Effective Approach for Unsupervised Graph Domain Adaptation[J]. arXiv preprint arXiv:2412.11654, 2024.

**Experimental Designs Or Analyses:**

Although the authors use 5 different datasets in the experiments, the authors use accuracy for all the datasets, which is not reasonable.

As for the findings, in Figure 2, the authors get the results by using GCN with standard unsupervised GDA setting as evaluation model. However, it is unclear what is the standard unsupervised GDA setting, i.e., MMD or adversarial training.

**Methods And Evaluation Criteria:**

The evaluation metric is not accurate. As different datasets have different properties, the accuracy (ACC) metric cannot reflect the model’s true performance on these datasets. For instance, MAG dataset’s labels are highly skewed, and macro-F1 score should be used as the evaluation metric. Twitch dataset contains two labels and should use AUC as the evaluation metric.

The model’s design also lacks justification and is contradictive to the graph spectral theory.

**Other Comments Or Suggestions:**

There are too many typos in the paper:

1.	Lines 29 to 31, as shown in Figure 1(a), “…ACM3 and ACM4…”. It should be Figure 1(b).

2.	Lines 143 to 144, there are two “where”.

3.	Lines 190 to 191, “heterophilc” should be heterophilic.

4.	Lin3 239, 257, 271, I don’t understand what $H_LH^{l-1}W_L^{l-1}$, $H_FH^{l-1}W_F^{l-1}$ and $H_HH^{l-1}W_H^{l-1}$ mean.

5.	In Appendix table 6, the dataset should be Twitch not Twitter.

**Other Strengths And Weaknesses:**

Pros:

1.	Graph domain adaptation could be a promising application, which helps alleviate distribution shift problem.

2.	Experiments are conducted on 5 public datasets including both small- and large-scale datasets.

3.	Ablation studies and theoretical analyses are given to show the effectiveness of the proposed model.

Cons:

1.	The design of the model needs more justification. The authors propose to utilize three filters to learn graph signals, which is defined as homophilic filter, full-pass filter and heterophilic filter. Based on the Equations (8), (9) and (10), the authors use $AX$, $IX$ and $LX$ to represent the homophilic, full and heterophilic signals in graph. Why Laplacian matrix $L$ can represent the heterophilic signals, which is contradictive to the graph spectral theory.

2.	The evaluation metric is not accurate. As different datasets have different properties, the accuracy (ACC) metric cannot reflect the model’s true performance on these datasets. For instance, MAG dataset’s labels are highly skewed, and macro-F1 score should be used as the evaluation metric. Twitch dataset contains two labels and should use AUC as the evaluation metric.

3.	The key baselines are missing. More recent baselines should be discussed and compared. Please refer to the reference below [1,2,3].

4.	It is not clear how the proposed model enhances the graph homophily. More justification should be given.

5.	More ablation studies should be given to show the necessary to adding the representations of $Z_L$, $Z_H$ and $Z_F$.

Reference:

[1] Huang R, Xu J, Jiang X, et al. Can Modifying Data Address Graph Domain Adaptation?[C]//Proceedings of the 30th ACM SIGKDD Conference on Knowledge Discovery and Data Mining. 2024: 1131-1142.

[2] Liu M, Fang Z, Zhang Z, et al. Rethinking propagation for unsupervised graph domain adaptation[C]//Proceedings of the AAAI Conference on Artificial Intelligence. 2024, 38(12): 13963-13971.

[3] Chen W, Ye G, Wang Y, et al. Smoothness Really Matters: A Simple yet Effective Approach for Unsupervised Graph Domain Adaptation[J]. arXiv preprint arXiv:2412.11654, 2024.

**Questions For Authors:**

Please refer to the weakness part.

**Relation To Broader Scientific Literature:**

The key ideas of combining different types of graph signals are not new, which is like multi-view or multi-channel graph convolutional network.

Wang X, Zhu M, Bo D, et al. Am-gcn: Adaptive multi-channel graph convolutional networks[C]//Proceedings of the 26th ACM SIGKDD International conference on knowledge discovery & data mining. 2020: 1243-1253.

**Theoretical Claims:**

I did not carefully check the full proofs in the Appendix due to limited time.

---

> ### Author Rebuttal · Authors · 2025-04-01
>
> Thank you for your appreciated feedback. Below, we address the concerns and questions raised in the weaknesses section. Please feel free to reach out if further clarification is required.
>
> # Q1
>
> **Justification:** Our model is designed to separately process graph signals with different levels of homophily. Specifically, we use $AX$ to extract homophilic signals, $IX$ to obtain full-pass signals, and $LX$ to capture heterophilic signals. Furthermore, this design aligns with our theoretical analysis, which shows that the entire model can be optimized by minimizing the distributional shifts of ${D_{\text{KL}}(A^S X^S | A^T X^T)}$, ${D_{\text{KL}}(X^S | X^T)}$, and ${D_{\text{KL}}(L^S X^S | L^T X^T)}$.
>
> **Laplacian matrix $L$:** Regarding $L$, There might be some misunderstanding. In fact, the graph **Laplacian matrix can be viewed as a high-pass filter that obtains heterophilic signals**—a perspective that has been adopted in several existing works [1, 2, 3] (e.g., in Section2.1 of [1], it says "To address the heterophily challenge, high-pass(HP) filter $L$ is often used to replace low-pass(LP) filter $A$"). To the best of our knowledge, we are unaware of any graph theory that contradicts utilizing a Laplacian matrix to obtain heterophilic signals. We are pleased to provide more clarification and havea more detailed discussion with you regarding this matter if you still have concerns.
>
> [1] Luan S, Hua C, Xu M, et al. When do graph neural networks help with node classification? Investigating the homophily principle on node distinguishability. Advances in Neural Information Processing Systems, 2023.
>
> [2] Luan S, Hua C, Lu Q, et al. Revisiting heterophily for graph neural networks. Advances in neural information processing systems, 2022.
>
> [3] Li B, Pan E, Kang Z. Pc-conv: Unifying homophily and heterophily with two-fold filtering. Proceedings of the AAAI conference on artificial intelligence. 2024.
>
> # Q2
>
> Thanks for your constructive comment! Following your suggestion, we also conducted additional experiments, evaluating **MAG** using macro-F1 and **Twitch** using AUC to address your concern. Due to word limitations, we have provided visualizations of these tables in the supplementary material, accessible via the following link [MAG](https://files.catbox.moe/e1ppdo.png) and [Twitch](https://files.catbox.moe/rw69ik.png).
>
> # Q3
>
> We acknowledge the importance of comparing our approach with recent advancements, including TDSS, A2GNN, GraphAlign [1*,2*,3*]. We report the performance of HGDA and the baseline methods on the [Airport, ACM, Blog](https://files.catbox.moe/hnbk9e.png) and [Citation](https://files.catbox.moe/or0weu.png) datasets. For the results on **MAG** and **Twitch**, please refer to our response to **Q2**. Our results demonstrate that while their method performs well, our approach exhibits **outperformance**, highlighting the importance of minimizing homophily shift. In future version, we will include these baseline methods along with additional evaluation metrics.
>
> # Q4
>
> Thank you for your valuable comment. We believe there might be a slight misunderstanding regarding the notion of "enhancing graph homophily" in the context of our work. Our paper does not aim to improve or increase the inherent homophily of the graph itself. Instead, our focus is on highlighting the homophily shift that occurs between the source and target domains in graph domain adaptation (GDA), and proposing a method to mitigate this shift. We will further revise our paper to clarify this point. Regarding the justification on this point, we provide [experimental results](https://files.catbox.moe/egsaqz.png) that report the classification accuracy on target graph subgroups with varying levels of homophily. The results show that **$HGDA_L$** performs best in subgroups with high homophily, **$HGDA_F$** performs best in subgroups with intermediate homophily, and **$HGDA_H$** performs best in subgroups with low homophily. These findings underscore the effectiveness of our method, which employs a combination of filters to mitigate homophily discrepancies at various levels.
>
> # Q5
>
> Regarding the concern about **$Z_L$**, **$Z_F$**, and **$Z_H$**, we would like to clarify that these embeddings are obtained by applying homophilic, full-pass, and heterophilic filters, respectively. In **Table 1**, **Table 2**, and **Table 3**, the variants **$HGDA_L$**, **$HGDA_F$**, and **$HGDA_H$** exclusively utilize **$Z_L$**, **$Z_F$**, and **$Z_H$**, respectively. The results of the main experiments show that each variant performs differently across datasets, which we attribute to the intrinsic homophily distribution characteristics of each dataset. The effects of **$Z_L$**, **$Z_F$**, and **$Z_H$** are further demonstrated by the experiments discussed in **Q4**.
>
> For clarification, the GCN baseline under the standard unsupervised GDA setting employs a two-layer GCN architecture combined with an MMD loss for domain alignment.

---

> > ### Comment · Reviewer_oEYM · 2025-04-07
> >
> > Dear authors,
> >
> > Thank you for your rebuttal. However, it seems that the links you provided are no longer valid. Could you please check and share the updated links?

---

> > > ### Author Response · Authors · 2025-04-08
> > >
> > > We apologize for any inconvenience and are pleased to provide valid links to clarify our rebuttal. Specifically, we have included links corresponding to each question, hosted on Google Docs, Anonymous GitHub, and imgbb, which can be found in Q2, Q3, and Q4 of the rebuttal, respectively.
> > >
> > > # Q2
> > >
> > > Google Docs: [MAG](https://docs.google.com/document/d/1f04LxgLsOYilN6iBnTJZhkbQvscgBw1xxrUIE80SDXc/edit?usp=sharing) [Twitch](https://docs.google.com/document/d/1qY1O26L7pm8wuGEZ6vPToSONZD_-AJsKLJkMRt05-PA/edit?usp=sharing)
> > >
> > > Anonymous GitHub: [MAG](https://anonymous.4open.science/r/ICMLrebuttal-B40A/MAG_%20F1.png) [Twitch](https://anonymous.4open.science/r/ICMLrebuttal-B40A/Twitch_auc.png)
> > >
> > > imgbb: [MAG](https://ibb.co/qY94XPdG) [Twitch](https://ibb.co/hRkXxc3p)
> > >
> > > # Q3
> > >
> > > Google Docs: [Airport, ACM, Blog](https://docs.google.com/document/d/192-_ziiSFqO0J9ct9kgwFWAAqJAXlGo_2ZEjdBmPl7U/edit?usp=sharing) [Citation](https://docs.google.com/document/d/14tlIaq5X6SebK_eh-NMvYNssfBLjboH0bCwUH29n5TQ/edit?usp=sharing)
> > >
> > > Anonymous GitHub: [Airport, ACM, Blog](https://anonymous.4open.science/r/ICMLrebuttal-B40A/Airport,%20ACM,%20Blog.png) [Citation](https://anonymous.4open.science/r/ICMLrebuttal-B40A/Citation.png)
> > >
> > > imgbb: [Airport, ACM, Blog](https://ibb.co/jXr63Kk) [Citation](https://ibb.co/wZdjB7NR)
> > >
> > > # Q4
> > >
> > > Google Docs: [experiment results](https://docs.google.com/document/d/1l_IBf_05Ipy8UED84cpLds5hT2Mp-LlIMd1e8DoTQZM/edit?usp=sharing)
> > >
> > > Anonymous GitHub: [experiment results](https://anonymous.4open.science/r/ICMLrebuttal-B40A/experiment%20results%20homophily%20ratio.png)
> > >
> > > imgbb: [experiment results](https://ibb.co/HT5frDtq)
> > >
> > > We would be grateful if this could improve your understanding of our works. We would be pleased to include these HGDA results on these baseline methods, additional evaluation metrics, and other improvements in our future revision.

---

### Official Review · Reviewer_6Xce · 2025-03-13

**Overall Recommendation:** 4

**Summary:**

This paper proposes a novel Graph Domain Adaptation algorithm which solves graph homophily disparity for effective domain alignment. It shows that homophily distribution shifts exist wildly in GDA datasets and could damage GDA performance in both empirically and theoretically ways. Inspired by theoretical results, it also provides a method to mitigate this discrepancy through cross-channel homophily alignment (HGDA).

**Claims And Evidence:**

The paper claims that homophilic ratio divergence exhibits a negative correlation with the classification accuracy of target graph nodes. It provides empirical results in Figure 2, which contain target node classicifaction accuracy percent difference and homophily distribution shift in each corresponding subgroups.

**Essential References Not Discussed:**

To my knowledge, the paper discusses the most essential references.

**Experimental Designs Or Analyses:**

I check the validity of experiments, consisting of performance comparison, ablation study, and hyper-parameter and model efficient experiment analysis.

**Methods And Evaluation Criteria:**

While most aspects of the evaluation are comprehensive, with performance reported over eight recent baselines on six datasets, the evaluation benchmark is partially limited. For benchmark Twitch [1], Russia (RU) and Spain (ES) are not involved. I also want to see the HGDA performance in ogbn-arxiv[2], which involves temporal discrepancy with different publication years.

[1] Liu M, Fang Z, Zhang Z, et al. Rethinking propagation for unsupervised graph domain adaptation[C]. AAAI 2024

[2] Liu M, Zhang Z, Tang J, et al. Revisiting, Benchmarking and Understanding Unsupervised Graph Domain Adaptation[J]. NeurIPS 2024

**Other Comments Or Suggestions:**

In Table 1, a vertical line between the Airport and ACM datasets appears to be missing.

**Other Strengths And Weaknesses:**

Strengths:

1.This paper presents a well-founded study supported by both theoretical analysis and experimental evidence.

2.The method proposed in this paper is grounded in meaningful theoretical research.

Weaknesses:

1.The experimental evaluation in this paper is somewhat limited. As noted in the Evaluation Criteria, a more comprehensive experiment across additional datasets would strengthen the study and improve its validity.

2.The dotted line in the two sub-graphs of Figure 1 represents the overall graph node homophily ratio. However, the paper does not clearly explain the method used to calculate this value.

3.Can the authors clarify the impact of the three proposed alignment modules, namely ${\text{KL}}(Z_L^S \| Z_L^T)$, ${\text{KL}}(Z_H^S \| Z_H^T)$, and ${\text{KL}}(Z_F^S \| Z_F^T)$, on the classification performance of nodes with different homophily ratios? Intuitively, these modules should play distinct roles in the alignment process.

4.Why is ${D_{\text{KL}}(P_S^H \| P_T^H)} $, one of the terms in Theorem 1, considered a fixed value? Additionally, why is it not subject to optimization?

**Questions For Authors:**

Additional experiments, particularly on the ogbn-arxiv dataset, would help validate HGDA's performance in addressing temporal discrepancies. Furthermore, a running time analysis and comparisons with a broader set of baseline models are needed to strengthen the evaluation. Other questions refer to other strengths and weaknesses.

**Relation To Broader Scientific Literature:**

This paper provides a novel view on GDA tasks. Its method has theoretical guarantees. Although SA-GDA[1] is partially similar to HGDA in utilizing graph signal in spectral space, I recommend that the authors discuss their differences with that paper in related work.

**Theoretical Claims:**

This paper theoretically justifies the impact of a homophily distribution shift on GDA and demonstrates that this discrepancy can be mitigated by addressing the homophilic and heterophilic signals.

---

> ### Author Rebuttal · Authors · 2025-04-01
>
> Thank you for your constructive feedback! Below, we address the concerns and questions raised in the weaknesses section. Please feel free to reach out if further clarification is required.
>
> # Q1
>
> Thanks for your constructive comment! Following your suggestion, we also conducted [additional experiments](https://files.catbox.moe/ewkc85.png), evaluating HGDA performance on **ogbn-arxiv** dataset. Specifically, we report the performance of HGDA on three tasks. As for ogbn-arxivl networks, we choose the year to separate networks, which are collected from **1950-2016(50-16), 2016-2018(16-18), and 2018-2020(18-20)**. As for Twitch **RU and ES**, we are pleased to provide their experiment, which is as [follows link](https://files.catbox.moe/9hxty1.png). Our results demonstrate that while their method performs well, our approach exhibits **outperformance**, highlighting the importance of minimizing homophily shift. In future versions, we will include these baseline methods along with additional evaluation metrics.
>
> # Q2
>
> We apologize for any misunderstanding caused by the previous lack of clarity. To clarify, overall graph node homophily ratio is $\; \int_{0}^{1} v \,\cdot\, H^v_{\text{node}} \,\mathrm{d}v.$
>
> which actually is our **Definition 1 (Graph-Level Node Heterophily Distribution)** in **Section3.3**. This method computes the homophily distribution across the entire graph. As shown in **Fig. 1**, in the Airport dataset, while the overall graph-level node heterophily distributions of the BRAZIL and EUROPE subgraphs are relatively similar, significant differences exist in the local homophily subgroups. These observations highlight the need for our model to handle homophily shifts effectively at different levels.
>
> # Q3
>
> To address your concern, we provide [experimental results](https://files.catbox.moe/egsaqz.png) that report the classification accuracy on target graph subgroups with varying levels of homophily. We also provide [synthetic experiments](https://files.catbox.moe/9odyw2.png) to validate the effectiveness of our three filter pairs in addressing varying levels of homophily shift, which detail can be found in **Reviewer BNmW** in **Q4**. The results show that **$HGDA_L$** performs best in subgroups with high homophily, **$HGDA_F$** performs best in subgroups with intermediate homophily, and **$HGDA_H$** performs best in subgroups with low homophily. These findings support the effectiveness of our method in aligning graph signals across different levels of homophily. Therefore, we can conclude that these three modules provide different roles in HGDA.
>
> # Q4
>
> Regarding **Definition 1 (Graph-Level Node Heterophily Distribution)**, $P_G^H$ captures structural information inherent to the graph, as observed in nature [1, 2]. Consequently, unless the graph topology itself is directly altered, $P_G^H$​ remains a fixed quantity. As a result, ${D_{\text{KL}}(P_S^H \| P_T^H)}$ is also an intrinsic and fixed value. Following your suggestion, we will address these concerns and provide clarification in a future version of the paper.
>
> [1] Pan E, Kang Z. Beyond homophily: Reconstructing structure for graph-agnostic clustering. International conference on machine learning. , 2023.
>
> [2] Xie X, Chen W, Kang Z. Robust graph structure learning under heterophily. Neural Networks, 2025.
>
> Thank you for your comment regarding Table line problems. We apologize for this and have carefully reviewed and corrected them throughout the paper in the future version.

---

> > ### Comment · Reviewer_6Xce · 2025-04-04
> >
> > I appreciate the rebuttal, and it addresses my concerns regarding the usefulness of the results. After seeing the discussion in the rebuttal, I am leaning toward accepting this paper. Please ensure the additional datasets HGDA results are also included in the final manuscript.

---

> > > ### Author Response · Authors · 2025-04-04
> > >
> > > Dear Reviewer,
> > > Thank you very much for your thoughtful and constructive feedback. We sincerely appreciate your time and effort in reviewing our work. We will include the HGDA results on the additional datasets, along with the other improvements, in our future revision. We are grateful for your positive evaluation, and we would truly appreciate it if you could consider reflecting this in your final score.
> > > Sincerely,
> > > The Authors

---

### Official Review · Reviewer_4CE1 · 2025-03-13

**Overall Recommendation:** 4

**Summary:**

This paper studies the problem of Graph Domain Adaptation problem through analysis homophily shift. This study reveals that homophily distribution shift negatively influences target domain accuracy in an empirical study. Empirical study reveals that homophily discrepancy exists in many benchmarks and provides an essential role in GDA. Through theoretical analysis using the PAC-Bayes framework, the authors prove that the domain shift is bounded by graph homophily distribution shift. Moreover, their theoretical analysis shows that homophily shift can be mitigated through aligning different signals.  The authors conducted comprehensive experiments to validate the algorithm's performance with a sufficient number of baseline methods compared.

**Claims And Evidence:**

Yes. The claims made in the submission are generaly supported by both theoretical and experimental evidence.

**Essential References Not Discussed:**

Some recent advancements in GDA might also need to cite, such as those discussed in [1].

[1] Zhang, Zhen, et al. "Aggregate to Adapt: Node-Centric Aggregation for Multi-Source-Free Graph Domain Adaptation.'' *The Web Conference*, 2025

**Experimental Designs Or Analyses:**

The exoerimental designs are generally sound whicn include the ablation study and parameter analysis.

**Methods And Evaluation Criteria:**

Yes. The proposed method make sense in reducing graph homophily discrepancy and the experimental evaluation appears reasonable and is conducted using widely accepted criteria.

**Other Comments Or Suggestions:**

Typos:
- Line 282 "Alignemnt"

- Line 98 "homophilc"

**Other Strengths And Weaknesses:**

Strengths:

- This paper highlights the importance of homophily distribution discrepancy in GDA and empirically investigates its impact on GDA performance.

- This paper is technically sound and novel. In theoretical analysis part, it demonstrates that the heterophily distribution shift between the source and target graphs can mitigate through homophilc signal, graph attribute signal, and heterophilic signal.

- HGDA method is theoretically motivated and aligns the three graph signals using KL divergence, ensuring consistency with the theoretical findings.

- The experiments are convincing and the experiment details are complete with detailed experiment description in the supplemental material.

Weaknesses:

- The explanation of the relationship between the proposed method and Theorem 1 in Section 4.2 could be clarified, particularly regarding their differing motivations.

- The paper should expand the discussion in the related work section to include connections to other studies that address structural shift in GDA. Since homophily is fundamentally a structural property, drawing parallels with existing approaches to structural shift could provide a broader contextual foundation for the study.

- Although the paper has generally sufficient experimental data set, it could benefit from additional experiments on other benchmarks for GDA’s target node classification task, e.g., ogbn-arxiv.

**Questions For Authors:**

See the weakness above.

**Relation To Broader Scientific Literature:**

The key contributions of the paper are related to the filed of graph domain adaptation and the homophily shift is of particular studied.

**Theoretical Claims:**

Yes. I have reviewed the proofs of the theoretical claims in the paper in the supplemental material particular for Theorem 1.

---

> ### Author Rebuttal · Authors · 2025-04-01
>
> Thank you for your feedback. We would also appreciate your agreement on our method's novelty and effectiveness. Below, we address the concerns and questions raised in the weaknesses section. Please feel free to reach out if further clarification is required.
>
> # Q1
>
> $ KL(Z_L^S \| Z_L^T) $ aligns the homophilic signal, corresponding to the term $ D_{\text{KL}}(A^S X^S \| A^T X^T) $. Similarly, $ KL(Z_H^S \| Z_H^T) $ directly aligns the graph attributes, corresponding to the term $ D_{\text{KL}}(X^S \| X^T) $. Finally, $ KL(Z_F^S \| Z_F^T) $ aligns the heterophilic signal, which is consistent with the term $ D_{\text{KL}}(L^S X^S \| L^T X^T) $ in **Theorem 1**. Specifically, this can be explained as follows. The term ${D_{\text{KL}}(A^S X^S | A^T X^T)}$ quantifies the divergence in the graph homophily signal, capturing how graph attributes—modulated by the adjacency matrices—differ between the source and target graphs. In contrast, ${D_{\text{KL}}(X^S | X^T)}$ measures the divergence in the distribution of graph attributes between the two domains. Lastly, ${D_{\text{KL}}(L^S X^S | L^T X^T)}$ quantifies the divergence in the graph heterophilic signal, where the attributes are modulated by the graph Laplacian matrix.
>
> # Q2
>
> Some early works have addressed graph homogeneity through reconstruction [1] to enhance graph homophily, our work does not directly modify the graph structure due to the computational complexity involved. Instead, **our method focuses on processing homophilic information at varying levels by grouping nodes accordingly**. Inspired by theoretical insights, the model employs low-pass, full-pass, and high-pass filters to capture homophilic signals at different levels. We are pleased to discuss these topics in the related work section of the future version.
>
> [1] Pan E, Kang Z. Beyond homophily: Reconstructing structure for graph-agnostic clustering. International conference on machine learning. , 2023.
>
> # Q3
>
> Thanks for your constructive comment! Following your suggestion, we also conducted [additional experiments](https://files.catbox.moe/ewkc85.png), evaluating HGDA performance on ogbn-arxiv dataset. Specifically, we report the performance of HGDA on three tasks. As for ogbn-arxiv networks, we choose years to separate networks, which are collected from **1950- 2016 (50- 16), 2016- 2018 (16- 18), and 2018- 2020 (18- 20)**. Our results demonstrate that while their method performs well, our approach exhibits **outperformance**, highlighting the importance of minimizing homophily shift. We will include these baseline methods and additional evaluation metrics in future versions.
>
> Thank you for your comment regarding typo problems. We apologize for this and have carefully reviewed and corrected them throughout the paper in the future version.

---

### Official Review · Reviewer_BNmW · 2025-03-13

**Overall Recommendation:** 3

**Summary:**

This paper investigates the graph domain adaptation (GDA) problem highlighting the importance of handling the shift across graph homophily distribution between the source and target graphs. They motivate the issue from both the empirical aspect and from theoretical analysis. Empirically, it has been observed that distinct proportion of different homophily subgroups across the source and target graph will impact the target performance. Theoretically, they show an error bound in terms of homophilic signal shift, heterophilic signal shift, node feature shift and node heterophily distribution shift. To handle the shifts, they propose the algorithm HGDA that align three types of signal shifts using KL divergence together with with source classification loss and target entropy loss. They evaluate the variants of HGDA with DA and GDA baselines on wide range of real world datasets.

**Claims And Evidence:**

**Strength:**
- Novel shift consideration: focusing on the homophily of subgroups and the entire distribution of node homophily is an interesting and valuable direction in addition to previous GDA works.
- Empirical justification: The figure 1/7 justify that this type of shift in node homophily exist in the real world datasets. Figure 2/8 try to demonstrate the empirical performance for subgroups with different homophily ratio against distinct homophily divergence.

**Question/Weakness:**
- It is good that you include Fig 5 as a comparison to Fig 2 after adopting the proposed method. It seems that after HGDA, we have a balance performance with different subgroups instead of related to the level of homophily divergence in fig 2. Can you elaborate more on why your method can handle different divergence level well?

**Essential References Not Discussed:**

This paper includes a wide range of relevant literatures but could elaborate more on how their problem and method compared to previous literature.

**Experimental Designs Or Analyses:**

**Strength:**
- Compared to many baselines and real world datasets
- Include some visualizations and analysis over parameters and different variants

**Weakness:**
- No indication of repeated experiments and no standard deviation included in the results
- Lack analysis in the three variants: For instance, in principle, $HGDA_F$ does not use graph information and should have similar results with DANN, but why it can have better results than graph-based methods and have comparable results with the other two variants?
- Better if you can provide some synthetic experiments or more detailed analysis over real datasets that detailing how HGDA handle different types/levels of shift

**Methods And Evaluation Criteria:**

**Strength:**
- The method itself is easy to follow and use
- The method design generally follows the motivation, empirical and theoretical analysis

**Weakness/Questions:**
- The method primarily focus on feature alignment which might be suboptimal in terms of graph data
- There is no specific and explicit handle that might target different levels of homophily divergence, i.e. the fourth term in the theoretical analysis. Although it appears to be an intrinsic graph parameter, but it might be able to handle empirically under GNN since it is a main part of your motivation saying this divergence causes performance degradation.
- Potentially lack control on how to determine the importance of homophilic/heterophilic alignment, should that depend on the distribution of the node homophily? Also, rather complicated loss for training.
- How this method can handle covariate shift, label shift and conditional structure shift that are previously discussed in GDA literatures? How you position homophily shift with previous discussed shifts and literature.

**Other Comments Or Suggestions:**

some typos: "Alignemnt"->"Alignment" in subtitle 4.2 and page 6 line 282

**Other Strengths And Weaknesses:**

Please refer to the above sections

**Questions For Authors:**

- Question regarding Fig.2: There seems to be a consistent gap between the two accuracy lines indicating two adaptation directions. Your plot shows that the accuracy only relates to the absolute gap in homophily distribution divergence but not with the directions. Can you explain why this is the case? Also, regarding the consistent gap between adaptations from two directions, if it is not attributed to the homophily distribution, what can be the potential cost? e.g. E & B has a large gap in (c) and A3 & A4 has a small gap.

- Question regarding training: How is the loss converge during training since we have a rather complex loss terms with many terms? Why choosing KL divergence in particular to align the distribution of filtered signals.

**Relation To Broader Scientific Literature:**

This paper helps bring up another issue that exist in GDA

**Theoretical Claims:**

The theoretical analysis tends to largely rely on results from previous works with limited contribution in novelty. Also, the analysis regarding KL divergence decomposition of feature distributions tends to be oversimplified and the bound seems to be not tight.

---

> ### Author Rebuttal · Authors · 2025-04-01
>
> Thank you for your feedback. We would also appreciate your agreement on our method's novelty and effectiveness. Below, we address the concerns and questions raised in the Claims And Evidence, Weaknesses, Theoretical Claims, and Questions For Authors section. Please feel free to reach out if further clarification is required.
>
> # Q1
>
> As shown in Fig. 5, HGDA achieves balanced performance across different subgroups by processing graph signals with varying levels of homophily through three specialized filters. Specifically, **$HGDA_L$** performs well in homophilic subgroups, **$HGDA_H$** excels in heterophilic subgroups, and **$HGDA_F$** is most effective in subgroups with intermediate homophily levels. The combined contributions of these components lead to an overall balanced performance.
>
> # Q2
>
> 1. We acknowledge that continued research into graph-structured data will further enhance the effectiveness of such approaches. However, HGDA method incorporates both homophilic and heterophilic filters that leverage the structural information of the graph.
>
> 2. Our motivation is to address subgroups with varying levels of homophily. To this end, we employ a homophilic filter to extract subgroups with **high homophily**, a full-pass filter for those with **middle homophily**, and a heterophilic filter for subgroups with **low homophily** [1]. This behavior can be observed in Fig. 5. As stated in **Theorem 1**, ${D_{\text{KL}}(P_S^H | P_T^H)}$ cannot be directly optimized as shown in **Definition1**. However, we can instead minimize the divergence of different homophily-level signals by optimizing ${D_{\text{KL}}(A^S X^S | A^T X^T)}$, ${D_{\text{KL}}(X^S | X^T)}$, and ${D_{\text{KL}}(L^S X^S | L^T X^T)}$, thereby alleviating homophily divergence.
>
> 3. In this paper, we focus on addressing the challenge of homophily shift. We acknowledge that covariate shift, label shift, and particularly conditional structure shift are also critical issues related to homophily in graph domain adaptation (GDA), which we plan to explore in future work.
>
>
> # Q2
>
> As noted in **Appendix B**, each experiment was repeated five times, and the reported results represent the average performance. Additionally, we will include the standard deviation of the results in future versions of the paper.
>
> # Q3
>
> The key difference between **$HGDA_F$** and **DANN** lies in their alignment strategies: $HGDA_F$ employs a KL divergence-based alignment loss, whereas DANN uses an adversarial loss. Moreover, DANN does not incorporate the pseudo-label classification loss on the target graph, which is included in our approach. These differences likely account for the performance gap observed between the two methods. Overall, the performance of different HGDA variants is related to the underlying distribution of the dataset. Specifically, in this [experiment](https://files.catbox.moe/egsaqz.png), **$HGDA_L$** and **$HGDA_F$** tend to perform better on datasets with a higher proportion of homophilic subgroups, while **$HGDA_H$** and **$HGDA_F$** perform better on datasets with a relatively higher degree of heterophily. In the meanwhile, we really appreciate your question, which gives us the opportunity to further improve our manuscript. Specifically, in light of your comments, we will revise **Section 5.3** to further clarify our motivation and avoid potential confusion.
>
> # Q4
>
> We provide [synthetic experiments](https://files.catbox.moe/9odyw2.png) to validate the effectiveness of our three filter pairs in addressing varying levels of homophily shift. Specifically, we randomly generated five sets of source and target graphs, each containing 300 nodes, where the homophily properties for each subgroup were varied in increments of 0.01. In the picture, the horizontal axis denotes the various homophily subgroups of target graphs, and the vertical axis indicates the performance of different HGDA variants across these homophily levels. The results indicate that $HGDA_L$ performs best in high-homophily scenarios, $HGDA_F$ excels in medium-homophily settings, and $HGDA_H$ is most effective in low-homophily contexts.
>
> # Q5
>
> We would like to further clarify that, as demonstrated in **Theorem 1**, the performance differences between source and target domain adaptation are influenced not only by component $\sqrt{D_{\text{KL}}(P_S^F \| P_T^F)}$, but also by component **$ L^\gamma_S(\phi)$**. Moreover, the effectiveness of **$L^\gamma_S(\phi)$** across different adaptation tasks also **depends on the source domain's empirical risk, which indicates an accuracy gap between two adaptation directions**.
>
> # Q6
>
> The use of KL divergence in our loss function is theoretically motivated. Additionally, incorporating these three loss terms does not significantly increase computational overhead. The overall computational complexity of HGDA remains controlled at **$O(N^2 \cdot d)$**.

---

> > ### Comment · Reviewer_BNmW · 2025-04-04
> >
> > Thank you for the response, it addressed some of my questions so I will raise my score to 3.

---

> > > ### Author Response · Authors · 2025-04-04
> > >
> > > Dear Reviewer,
> > > Thank you for your thoughtful feedback and for increasing the score of our manuscript. We sincerely appreciate your insightful questions, as addressing and clarifying them has significantly strengthened our paper.
> > > Sincerely,
> > > The Authors

---

### Decision · Program_Chairs · 2025-05-01

**Decision:**

Accept (poster)

**Comment:**

This paper tracks an important problem: graph domain adaptation. Given the graph data is more and more important in real applications, investigating the effect brought by distribution shifts is important. The main contribution of this paper is to find that homophily discrepancies exist in benchmarks and degrade GDA performance from both empirical and theoretical aspects. This is a new contribution to this field.

The reviewers had concerns regarding the technical novelty used in this paper. Indeed, the final methodology is not entirely new, as some high-level ideas already exist in the literature. However, given the contribution of this paper lies in the findings above, this paper still has enough contributions to the field. This is also agreed by reviewers who are all positive towards this paper at the end of the discussion.